# Automated determination of landslide locations after large trigger events: advantages and disadvantages compared to manual mapping

David G. Milledge[1], Dino G. Bellugi[2], Jack Watt[3], Alexander L. Densmore[3]

[1]School of Engineering, Newcastle University, Newcastle upon Tyne, UK
[2]Department of Geography, University of California, Berkeley, CA, USA
[3]Institute of Hazard, Risk, and Resilience and Department of Geography, Durham University, Durham, UK

*Correspondence to*: David G. Milledge (david.milledge@newcastle.ac.uk)

**Abstract.** Earthquakes in mountainous areas can trigger thousands of co-seismic landslides, causing significant damage,
hampering relief efforts, and rapidly redistributing sediment across the landscape. Efforts to understand the controls on these
landslides rely heavily on manually mapped landslide inventories, but these are costly and time-consuming to collect, and their
reproducibility is not typically well constrained. Here we develop a new automated landslide detection algorithm (ALDI) based
on pixel-wise NDVI differencing of Landsat time series within Google Earth Engine accounting for seasonality. We compare
classified inventories to manually mapped inventories from five recent earthquakes: 2005 Kashmir, 2007 Aisen, 2008
Wenchuan, 2010 Haiti, and 2015 Gorkha. We test the ability of ALDI to recover landslide locations (using ROC curves) and
landslide sizes (in terms of landslide area-frequency statistics). We find that ALDI more skilfully identifies landslide locations
than published inventories in 10 of 14 cases when ALDI is locally optimised, and in 8 of 14 cases both when ALDI is globally
optimised and in holdback testing. These results reflect both good performance of the automated approach but also surprisingly
poor performance of manual mapping, which has implications not only for how future classifiers are tested but also for the
interpretations that are based on these inventories. We find that manual mapping, which typically uses finer resolution imagery,
more skilfully captures the landslide area-frequency statistics, likely due to reductions in both censoring of individual small
landslides and amalgamation of landslide clusters relative to ALDI. We conclude that ALDI is a viable alternative to manual
mapping in terms of its ability to identify landslide-affected locations, but is less suitable for detecting small isolated landslides
or precise landslide geometry. Its fast run-time, cost-free image requirements and near-global coverage suggest the potential
to significantly improve the coverage and quantity of landslide inventories. Furthermore, its simplicity (pixel-wise analysis
only) and parsimony of inputs (optical imagery only) mean that considerable further improvement should be possible.

## 1 Introduction

Landslides are important as both agents of erosion and as a dangerous hazard (Marc et al., 2016; Froude and Petley, 2018).
Large earthquakes or rainstorms can trigger thousands of landslides, redistributing tonnes of rock over distances of hundreds
or thousands of metres within a few seconds (Li et al., 2014; Roback et al., 2018). These landslides can cause significant
damage, hamper relief efforts, and rapidly redistribute sediment across the landscape. Efforts to understand the drivers,
behaviour, and consequences of these landslides rely heavily on landslide inventories, in which landslide locations are mapped
either as points, pixels, or polygons, usually associated with one or more assumed trigger events. Landslide inventories are
important because they document the extent and impact of landslides in a region, informing disaster response and recovery
(Williams et al., 2018); they capture the distribution, properties, and (through predictive models) drivers of landslides (Guzzetti
et al., 2012, Tanyas et al., 2019); they can be used to train and evaluate models of landslide susceptibility, hazard, and risk
(Van Westen et al., 2006; Reichenbach et al., 2018); and they enable geophysical flux calculations central to the study of
landscape evolution and the global carbon cycle (e.g., Hilton et al., 2008; Marc et al., 2016, Dietrich et al., 2003).
Polygon-based and pixel-based inventories both capture information on the area affected by landslide movement. Polygon-
based inventories have the additional advantage that they can be analysed to yield distributions of landslide geometry (such as

area and shape), which is useful for understanding fluxes of material (Larsen et al., 2010) or impact forces, and distinguishing scars from runout areas (Marc et al., 2018).

Landslide inventories were traditionally generated from expensive and time-consuming site visits (e.g., Warburton et al., 2008), severely limiting the number of landslides that could be mapped and thus the scale of enquiry. However, they are now
increasingly collected remotely based on interpretation of satellite or aerial imagery, allowing the compilation of much larger datasets (e.g., Li et al., 2014; Roback et al., 2018).

Imagery provides an opportunity for rapid mapping over wide areas but is subject to some important limitations. For optical imagery, which depends on reflected solar energy reaching the sensor, cloud and shadow can obscure the ground surface. Active sensors, such as radar, that operate at wavelengths that are not reflected by cloud suffer from other issues (e.g., radar
layover and shadowing) and their images are only recently being incorporated into operational landslide mapping approaches (e.g., Konishi and Suga, 2018; Burrows et al., 2019; Aimaiti et al., 2019; Mondini et al., 2019). Images may not be available for the study area over the time window of interest, and - when they are available - they can be costly to acquire. In steep or high-relief topography, images can suffer severe geo-rectification errors (Williams et al., 2018), which is particularly problematic for landslide mapping because these are the areas of most interest. Imagery is becoming increasingly available
across a very wide range of spatial and spectral resolutions but there remains a trade-off between resolution and cost, with 10-30 m imagery freely available globally with a 14-day revisit time (e.g., Sentinel 2, Landsat 8) while sub-metric resolution data (e.g., Worldview, Pleiades) can be acquired on demand but at a cost of $10^1$-$10^4$ USD/km$^2$.

Landslides are typically identified in imagery either by automated classification, manual mapping, or some hybrid of the two. Manual mapping, although much faster than site visits, remains very time consuming over moderate to large areas (Galli et
al., 2008), particularly for co-seismic inventories, which can involve digitising $10^4$ to $10^5$ landslides (e.g., Xu et al., 2014; Harp et al., 2016). It also requires comparison of pre- and post-event images to identify change and to avoid conflation of landslides related to the trigger event with those occurring before or after the event (e.g., Hovius et al., 2011; Marc et al., 2015). Automated classification can considerably speed up this process but is complicated by other factors, including: the range of possible landslide sizes and geometries; the non-unique signatures of landslides relative to roads, buildings, or other features;
and the difficulty of excluding pre-existing landslides (Parker et al., 2011; Behling et al., 2014). Automated landslide classification has been demonstrated predominantly using high-resolution imagery and requires a high level of tuning, thus it is not necessarily transferrable from one region or event to another. Imagery can be combined with other sources of information (e.g., slope inclination from DEMs) to remove some false positives, where a location is incorrectly classified as a landslide (Parker et al., 2011). This can improve classifier performance but can also generate spurious correlation when interpreting the
results (e.g., landslide susceptibility with slope inclination). Some authors have adopted hybrid approaches; for example, Li et al. (2014) applied manual checking to the earlier automated mapping of Parker et al. (2011).

As a result of these issues, our database of landslide inventories is limited in number and biased towards the most spectacular trigger events. This point is most easily illustrated by examining earthquake-triggered landslide inventories since in this case the trigger event is generally very clearly identifiable in time and its footprint is well defined in space. Of the 326 earthquakes
known to have triggered landslides between 1976 and 2016, only 46 have published landslide maps (Tanyas et al., 2017). For 225 earthquakes the existence of co-seismic landslides was known from news reports and witness testimony (Marano et al., 2010), but no reliable quantitative or spatial landslide data are available (Tanyas et al., 2017). Many other earthquakes have likely triggered landslides, but these have gone unreported because they occurred out of human view. Between 1976 and 2016 there were ~6500 earthquakes sufficiently large (>$M_w$ 5), shallow (<25 km) and near to land (<25 km) to trigger landslides
(based on Marc et al., 2016). This suggests that the existing set of co-seismic landslide inventories is a small subset (<15%) of those earthquakes known to have triggered landslides and a tiny subset (<1%) of those likely to have triggered landslides.

To extend the number of landslide inventories requires a reduction in the cost of inventory collection, both in terms of imagery expense and mapping time. We hypothesise that recent improvements in satellite data management (e.g., data cubes) and

computing capabilities (e.g., cloud computing) have made it possible to collect automated landslide inventories of comparable quality to manual mapping, and at a fraction of the cost, due to reductions in both imagery cost and mapping time. Imagery cost could be reduced by using cheaper, lower resolution imagery, while mapping time could be reduced by using automated detection rather than manual mapping. However, these savings will only represent value for money if they can deliver inventories of comparable or superior quality to manual mapping.

Large amounts of freely-available optical imagery with near-global coverage have been generated by the Landsat and Sentinel programmes. Landsat has been running for more than 30 years (since the Landsat 4 launch in 1982), imaging the majority of the Earth's surface at a return time of c. 14 days and at 30 m spatial resolution through the visible and infra-red bands. Landsat received early attention as a source of imagery for manual landslide mapping (e.g., Sauchyn and Trench, 1978; Greenbaum et al., 1995) but has since been largely superseded by imagery with higher spatial resolution, which is often assumed to result in more precise inventories (e.g., Parker et al., 2011; Li et al., 2014; Roback et al., 2018). The recent HazMapper application of Scheip and Wegmann (2021) is a notable exception, and seeks to leverage the large volume of freely-available coarser resolution imagery to provide information on vegetation change that can be used to map a range of hazards including landslides. It is not clear, however, whether the long time series of coarser-resolution imagery that is now available contains as much usable information as individual images of finer resolution.

There have been some attempts at automated landslide detection from Landsat (e.g., Barlow et al., 2003; Martin and Franklin, 2005). However, manual mapping remains the most common approach to map landslides despite the time costs associated with it. Automated or hybrid approaches still need visual interpretation for calibration, sometimes over large areas (e.g., Đuric et al., 2017) and are typically compared to a manual map of landslides that is considered to represent the 'ground truth' (van Westen et al., 2006; Guzzetti et al., 2012; Pawłuszek et al., 2017; Bernard et al., 2021). There remains a perception in the landslide community that automated methods are neither necessarily more accurate (Guzzetti et al., 2012; Pawłuszek et al., 2017) nor less time consuming (Santangelo et al., 2015; Fan et al., 2019) than manual interpretation. Given the considerable investment of time and money involved in compiling an inventory, many researchers continue to generate inventories through manual mapping. It is therefore timely and useful to evaluate both automated classification and manual mapping against a common measure of performance.

Establishing the performance of an automated classifier against manual mapping requires both establishing the landslide characteristics that should be reproduced and establishing the quality of manual mapping with respect to these characteristics. This is typically done by comparing similarity between at least two independently-collected landslide inventories in terms of their overlap, or the similarity in their area-frequency distributions. Uncertainty in area-frequency distributions from manually-mapped landslide inventories has received considerable attention (e.g. Galli et al., 2008; Fan et al., 2019; Tanyas et al., 2019) but uncertainty in landslide spatial properties has received relatively little attention. However, the limited number of studies that do quantify landslide inventory error all suggest very weak spatial agreement between different manually-mapped landslide inventories. Ardizzone et al. (2002) found 34-42% overlap between three inventories for the same study area (i.e., 34-42% of the area classified as a landslide in one inventory was classified as a landslide in another). Galli et al. (2008) found 19-34% overlap for three different inventories and Fan et al. (2019) found 33-44% overlap for three inventories associated with the Wenchuan earthquake. Fan et al. (2019) also compared their own inventory to the three published inventories and found overlaps of similar magnitude (32-47%) with two inventories but a much closer agreement (82% overlap) with the third; however, they did not suggest a reason for this closer agreement. These low similarity figures suggest that caution is needed in assuming that any one inventory represents a 'ground truth'.

This research seeks to test our hypothesis that an automated detection algorithm applied to a time series of lower-resolution imagery can deliver inventories of comparable quality to those generated from manual mapping of higher-resolution imagery. We introduce a new approach to automated landslide detection using Landsat time series in Google Earth Engine (GEE). Our approach uses similar data and architecture to HazMapper but is focused on landslides in particular and uses an expectation of

long- and short-term change rather than a straight comparison of pre- and post-event composite images (Scheip and Wegmann, 2021). To account for uncertainty in the quality of manually-mapped inventories, we apply this approach to case studies where there are at least two pre-existing inventories. This allows direct comparison of the inventories that we create (in terms of both landslide location and size) with multiple uncertain manually-mapped inventories. The key question: can landslide location and size be reproduced more skilfully by our automated approach than by a second manual inventory?

## 2 Case study sites

We choose earthquake-triggered landslide detection to test our hypothesis because: 1) this type of trigger is well constrained in time and its footprint is well defined in space; and 2) there are several earthquake case studies for which at least two landslide inventories are available in order to assess the quality of manual mapping. We choose five earthquake case studies in which at least two landslide inventories have been published and where the authors attributed the landslides to the same trigger event (i.e., earthquake timing and epicentral location). The mapping times given below are each team's estimates of the total number of person-days taken to map the landslides in their inventory; this is reported in the metadata associated with that team's submissions to the USGS Science Base catalogue of landslide inventories (Science Base Community, 2021).

The 2005 Kashmir, Pakistan, earthquake triggered >2,900 landslides with a combined area of ~110 km$^2$ across an area of 4,000 km$^2$ (Basharat et al., 2016). The study area is primarily underlain by sedimentary rock, with a summer monsoon climate and seasonal snow on the highest peaks (note that the climate is drier than the 2015 Gorkha study site). Landslides associated with the earthquake were mapped by Sato et al. (2007; 2017), who estimated that they spent 60 days mapping the landslides using 2.5 m resolution SPOT 5 optical satellite imagery, and by Basharat et al. (2016; 2017) over 90 days using 2.5 m resolution SPOT 5 imagery and field reconnaissance. The inventories of Sato et al. (2007; 2017) and Basharat et al. (2016; 2017), hereafter referred to as Sato and Basharat respectively, contain 2,424 and 2,930 landslides respectively.

The 2007 Aisen Fjord, Chile, earthquake triggered >500 landslides with a combined area of ~17 km$^2$ across an area of 1,500 km$^2$ (Sepulveda et al., 2010). The study area is glacially carved valleys in volcanic rock and has a temperate climate with seasonal snow throughout and perennial snow at altitude. The associated co-seismic landslides were mapped by Sepulveda et al. (2010a; 2010b) over 120 days using Landsat images and field mapping, and by Gorum et al. (2014; 2017b) over 5 days using 5 m resolution SPOT 5 imagery. The inventories of Sepulveda et al. (2010) and Gorum et al. (2014; 2017b), hereafter referred to as Sepulveda and Gorum respectively, contain 538 and 517 landslides respectively.

The 2008 Wenchuan, China, earthquake triggered >190,000 landslides with a combined area of ~1000 km$^2$ across an area of 75,000 km$^2$ (Xu et al., 2014). The study area is primarily underlain by meta-igneous and sedimentary rock with a humid temperate climate and snow cover limited to the highest peaks. The associated co-seismic landslides were mapped by Li et al. (2014; 2017) over 300 days using high (3-10 m) resolution optical satellite images, and by Xu et al. (2014; 2017) over 1200 days using high (1-20 m) resolution satellite images. The inventories of Li et al. (2014; 2017) and Xu et al. (2014; 2017), hereafter referred to as Li and Xu respectively, contain 69,606 and 197,481 landslides respectively.

The 2010 Haiti earthquake triggered >20,000 landslides with a combined area of ~25 km$^2$ (Harp et al., 2016) across an area of ~4,000 km$^2$. The study area is characterised by steep but low relief valleys cut through sedimentary rock with a humid temperate climate in which snow is extremely rare and a land-use regime in which the vegetation is rapidly changing. The associated co-seismic landslides were mapped by Gorum et al. (2013; 2017a) over 40 days using GeoEye-2 and Worldview-2 (0.6-1 m resolution) satellite images, and by Harp et al. (2016; 2017) using 0.6 m resolution aerial photographs and field mapping. The inventories of Gorum et al. (2013; 2017a) and Harp et al. (2016; 2017), hereafter referred to as Gorum and Harp respectively, contain 4,490 and 23,567 landslides respectively.

The 2015 Gorkha, Nepal, earthquake triggered >24,000 landslides with a combined area of ~87 km$^2$ across an area of 20,000 km$^2$ (Roback et al., 2018). The study area is primarily sedimentary and metamorphic rock with seasonal snow at higher

elevation and perennial snow and ice at highest elevations. The climate ranges from humid temperate to alpine with a strong summer monsoon. The associated co-seismic landslides were mapped by Zhang et al. (2016, 2017) over 20 days using Gaofen 1 and 2 (1-5.8 m resolution) and Landsat satellite images; by Roback et al. (2017, 2018) using Worldview satellite images (0.5-2 m resolution); and by Watt (2016) using Landsat satellite images. The inventories of Roback et al. (2017, 2018), Zhang (2016, 2017) and Watt (2016), hereafter referred to as Roback, Zhang and Watt respectively, contain 24,915, 2,643 and 4,924 landslides respectively. The Watt (2016) mapping reported here was undertaken for a period of 60 days and involved comparing pan-sharpened false colour composites (red, green and near infra-red) derived from Landsat 8 images before and after the earthquake. Mapping was undertaken from multiple images to minimise occlusion by cloud, but all images were acquired within one year before and after the earthquake. The majority of the study area was mapped by a single person based on comparison of one pre- and two post-event images (from 13/3/2015, 1/6/2015, and 7/10/2015). This mapping was checked and supplemented by a second mapper using the same procedure to capture previously occluded areas using seven more Landsat 8 images. The registration errors in the Watt (2016) inventory were estimated from those associated with the underlying imagery from which the landslides were mapped. These Landsat 7 and 8 images were all geo-referenced to Level 1TP resulting in radial root mean square error of <12 m (USGS, 2019), which is less than the pan-sharpened pixel resolution (15 m). We were unable to find registration error estimates for the other landslide inventories examined here.

## 3 Methods

### 3.1 ALDI classifier: theory

The ALDI algorithm leverages the change in vegetation cover (and associated spectral signature of reflected light) caused by the removal of vegetation by landslides. The change in spectral signature is typically characterised by a change in the normalised difference vegetation index (NDVI; Tucker, 1979), defined as:

$$NDVI = \frac{R_n - R_r}{R_n + R_r} \tag{1}$$

where $R_n$ is spectral reflectance in the near infra-red band and $R_r$ is spectral reflectance in the red band (wavelengths in Table 1). The light reflected from landslide-affected pixels, whether they are within the scar or runout area, has a spectral signature associated with rock or sediment. This differs considerably from vegetation in terms of $R_n$ and $R_r$, resulting in extremely low NDVI values. We call the difference in NDVI before and after the trigger event $dV$, which is bounded by [-1, 1] and should be negative for landslide pixels associated with the event. This is not in itself a novel approach and is similar to other NDVI differencing approaches (e.g. Behling et al., 2014; 2016; Marc et al., 2019; Scheip and Wegmann, 2021).

In addition, vegetation that is disturbed by landslides regrows slowly - over timescales of months to years (Restrepo et al., 2009). Thus, for landslide-affected pixels NDVI should not only reduce after the trigger event but also stay low for an extended period (at least one year, depending on climate and seasonality as well as the timing of the earthquake). Therefore, we examine a time series of post-event images to calculate a time-averaged post-event NDVI, which we call $V_{post}$, which is bounded by [0, 1] and which should be low for landslide pixels associated with the trigger event.

Averaging over a time series of images has the additional advantage that it enables robust estimates of both $dV$ and $V_{post}$ even for NDVI time series that are both patchy and noisy. The time series are patchy because cloud cover occludes the ground for some pixels on some days; this cloud can be removed using filtering algorithms (e.g., Irish, 2000; Goodwin et al., 2013) but this leaves a gap in the series. The timing and number of these gaps vary from pixel to pixel, making comparison of NDVI for particular dates or images problematic. The time series are noisy because atmospheric conditions alter both incoming radiation (e.g., cloud shadow) and that received by the sensor, and because ground surface (and especially vegetation) properties will vary over time both periodically (e.g., due to seasonal vegetation growth and harvesting) and randomly (e.g., due to leaf orientation).

Since we expect NDVI to be noisy, we seek a third metric to identify whether there is a shift in NDVI in the presence of broadly consistent seasonal variations and random noise in NDVI. For this we take the difference in NDVI across monthly bins to account for the seasonal component, then quantify the shift in NDVI since the trigger event. For the shift to be indicative of real change it should be considerably larger than the noise present in the NDVI signal. Thus, we express the NDVI shift relative to the noise for each pixel as.

$$t = \sqrt{n}\frac{dV}{S_v} \tag{2}$$

where $n$ is the sample size (12 for monthly bins), $dV$ is the mean of the monthly NDVI differences, and $S_v$ is the standard deviation of the monthly NDVI differences. We then normalise by mapping $t$ onto the cumulative Student's t distribution to generate $P_t$, the likelihood that the pre- and post-event NDVIs are drawn from different distributions:

$$P_t = I_{\frac{(n-1)}{n-1+t^2}}\left(\frac{n-1}{2},\frac{1}{2}\right) \tag{3}$$

where $I_x(a,b)$ is the regularized incomplete beta function. While this is equivalent to a paired t-test, the results cannot be interpreted as formal probabilities, as the distribution of $dV$ may not be Gaussian. Rather they represent an index of change relative to expected variability which is bounded by [0, 1]. $P_t$ should be high for landslide pixels associated with the trigger event. High $P_t$ could also result from other events that reduce the coverage or vigour of vegetation, particularly if this involves complete removal (e.g., fire or logging). However, seasonal vegetation changes should be accounted for by examining monthly differences, while episodic events should only be noticeable when: 1) their timing is coincident with the earthquake and 2) their effect persists over more than one year.

Although low NDVI is effective for identifying the absence of vegetation, it does not uniquely identify landslides since a range of other surfaces generate similar signatures, particularly snow and cloud. Cloud cover varies from one image to another, and we thus seek to remove cloud-affected pixels from both the pre- and post-event time series. Cloud can be identified based on its spectral signature, with different types resulting in different signatures. The 'Landsat simple cloudscore' function within Google Earth Engine returns the minimum of a set of five cloudiness indices using Equations 4a-f and parameters in Table 2 (Earth Engine, 2021). Each index reflects an expectation about cloud reflectance and temperature: they should be reasonably bright in the blue band ($CI_b$), in all visible bands ($CI_v$), and in all infra-red bands ($CI_{ir}$); and they should be reasonably cool in the thermal infra-red band ($CI_{Temp}$); but they should not be snow ($CI_{NDSI}$):

$$CI_b = \frac{R_b - R_{bmin}}{R_{bmax} - R_{bmin}} \tag{4a}$$

$$CI_v = \frac{(R_r + R_g + R_b) - R_{vmin}}{R_{vmax} - R_{vmin}} \tag{4b}$$

$$CI_{ir} = \frac{(R_n + R_{s1} + R_{s2}) - R_{irmin}}{R_{irmax} - R_{irmin}} \tag{4c}$$

$$CI_{Temp} = 1 - \frac{R_t - R_{tmin}}{R_{tmax} - R_{tmin}} \tag{4d}$$

$$CI_{NDSI} = 1 - \frac{NDSI - NDSI_{min}}{NDSI_{max} - NDSI_{min}} \tag{4e}$$

$$CI = \min\left(CI_b, CI_v, CI_{ir}, CI_{Temp}, CI_{NDSI}\right) \tag{4f}$$

where $R_g$, and $R_b$, are the spectral reflectances from the red and blue bands; $R_{s1}$ and $R_{s2}$ are those from the first and second shortwave infra-red bands; and $R_t$ is that from the thermal infra-red band (the only band used here with a coarser 60 m resolution). The parameters with min and max subscripts (e.g., $R_{bmin}$ and $R_{bmax}$ for the red band) in Equation 4 are minimum and maximum values used to normalise pixel reflectances, their values are given in Table 2. NDSI is the normalised difference snow index:

$$NDSI = \frac{R_g - R_s}{R_g + R_s} \tag{5}$$

This index is also used within ALDI outside the Landsat simple cloudscore function to identify pixels where persistent snow cover could result in misleading statistics. Where pixels remain snow-covered for periods of several weeks or months, we

cannot retain sufficient observations to calculate stable statistics from these pixels. Instead, we identify pixels with persistent snow cover based on time-averaged NDSI and censor them from the analysis.

We define the Automated Landslide Detection Index (ALDI) as the product of the three parameters defined above. While this
formulation is entirely arbitrary, it has the advantage of allowing the index to take a minimum value of zero (indicating negligible probability that the images reflect a landslide at that location) if any of the individual terms is zero. Because we have no *a priori* knowledge of the relative importance of each parameter in determining the landslide signature, we assume a power-functional form with empirical exponents α, β and λ:

$$ALADIN = \begin{cases} (-dV)^\alpha \left(1 - V_{post}\right)^\beta P_t^\lambda, & if\ S_{post} > T_{snow}\ |\ dV < 0 \\ 0, & otherwise \end{cases} \tag{6}$$

where $S_{post}$ is the mean post-earthquake NDSI and $T_{snow}$ is a threshold value for NDSI, chosen to identify persistent snow cover. The likelihood that a pixel is landslide-affected increases monotonically with the ALDI output value, which has upper and lower bounds of 0 and 1 respectively. Landslide pixels should be characterised by negative $dV$, indicating vegetation removal; low $V_{post}$, indicating a lack of vegetation after the earthquake; and high $P_t$, due to a distinguishable shift in post-event NDVI distributions relative to the pre-event distributions. The likelihood that a pixel contains a landslide should increase with $P_t$ and
decrease with $dV$ and $V_{post}$. We exclude snow-dominated pixels where $S_{post}$ exceeds a threshold $T_{snow}$, as well as pixels where median post-earthquake NDVI exceeds that pre-earthquake (i.e., positive $dV$).

The empirical exponents α, β and λ can be expressed in terms of one parameter (α) and two ratios (α:β and α:λ) because:

$$\beta = \alpha \frac{1}{\alpha:\beta} \ and \ \lambda = \alpha \frac{1}{\alpha:\lambda} \tag{7}$$

Substituting these terms into Equation 6:

$$ALADIN = \begin{cases} (-dV)^\alpha \left(1 - V_{post}\right)^{\alpha \frac{1}{\alpha:\beta}} P_t^{\alpha \frac{1}{\alpha:\lambda}}, & if\ S_{post} > T_{snow}\ |\ dV < 0 \\ 0, & otherwise \end{cases} \tag{8}$$

then taking logarithms of both sides clarifies the role of the ratio parameters:

$$log(ALADIN) = \alpha \left( log(-dV) + \frac{1}{\alpha:\beta} log\left(1 - V_{post}\right) + \frac{1}{\alpha:\lambda} \log(P_t) \right) \tag{9}$$

Since $dV$, $V_{post}$ and $P_t$ are all ≤ 1 (thus their logarithms are negative), and larger values of the ratio parameters (α:β and α:λ) result in smaller powers for their respective layers ($V_{post}$ and $P_t$), therefore large α:β ratios result in a stronger influence of $V_{post}$
on ALDI, large α:λ ratios result in the same for $P_t$, and when both α:β and α:λ are small $dV$ dominates. These ratios are more informative than the raw parameters because it is the relationship between exponents rather than the exponents themselves which define the relative role of the different ALDI components (i.e., equal but high values of α, β and λ result in the same ALDI classification pattern as equal but low values).

### 3.2 ALDI classifier implementation and data pre-processing

We implement ALDI and perform all pre-processing steps within Google Earth Engine (GEE; Gorelick et al., 2017) because: 1) it hosts an extensive Landsat archive and provides efficient access to large volumes of freely-available satellite data; 2) it provides both a toolkit of pre-compiled algorithms for image processing and cloud computing resources to run these algorithms; and 3) it is an open access platform so that both the data and the algorithms used here are widely accessible and reproducible (source code available in Supplementary Information).

The objective of pre-processing is to generate four layers: $dV$, the change in NDVI before and after the trigger event; $V_{post}$, the time-averaged post-event NDVI; $S_{post}$, the post-event NDSI; and $P_t$, the likelihood that pre- and post-event NDVIs are drawn from different distributions. These layers should synthesise the time series of available imagery from multiple sensors minimising bias due to the sensor, the influence of clouds, and seasonal vegetation changes.

We use time series of NDVI calculated from Landsat 5, 7 and 8 imagery following 'top of atmosphere' correction (Chandler
et al., 2009) to adjust for radiometric variations due to solar illumination geometry (angle and distance to Sun) and sensor

specific gains and offsets. Sentinel 2 data would offer additional gains in terms of both spatial and temporal resolution of data but are not available for any of our case study events and thus cannot yet be evaluated within the same framework. Landsat 8 sensors aggregate red and near infra-red reflectance over slightly different frequency bands to Landsat 5 and 7, but their central frequencies vary by <4% between sensors and by >20% between bands (Table 1). To ensure satisfactory image-to-image

registration for time series analysis, we use only images which have been both georeferenced to ground control points and terrain corrected (i.e. Level 1TP) and thus have ≤12 m radial root mean square error (RMSE) in >90% of cases (USGS, 2019). The time series is split into two 'stacks' of images, those before the trigger event and those after it (Figure 1b). The duration of these time series (and thus length of stacks) reflects a trade-off between shorter durations, which limit the sample size, and longer durations, which include landscape changes unrelated to the earthquake. We remove 'cloudy' pixels from each stack

using the GEE simple cloud score exceeding a tuneable threshold ($T_{cloud}$) where stricter thresholds remove more cloudy pixels but also incorrectly remove more cloud-free false positives (Earth Engine, 2018). The number of images in each stack is controlled by the stack lengths and cloud threshold, introducing three tuneable parameters to be calibrated. These parameters are found using the calibration process described in Section 3.4 rather than by considering the physical processes that characterise the possible evolution of the time series.

To account for seasonal vegetation change, NDVI values for each pixel in the pre- and post-earthquake stacks are extracted as a time series (Figure 1a) and binned based on the month in which the image was acquired. Monthly bins are used since they are generally long enough to contain data in every bin (even after removal of cloudy pixels) but short enough to capture annual seasonality (e.g., Figure 1a). Monthly bins result in four images per bin per year on average, and thus empty bins are very unlikely except for month-location pairs that are characterised by extreme cloudiness (such as Nepal in July; see Wilson et al.,

2016). Monthly bins that are empty in either pre- or post-earthquake period are not used in the subsequent analysis, with calculations for that pixel performed using the remaining monthly bins. We calculate median NDVI for each monthly bin, choosing median rather than mean since it is less sensitive to skew and to extreme values (Figure 1c). We difference the monthly median values prior to and after the trigger event, generating a distribution of differences (Figure 1c). From that distribution, we calculate the mean monthly NDVI difference, $dV$ and evaluate the likelihood that the mean monthly NDVI

difference differs significantly from zero using a pairwise t-test to calculate $P_t$. We take the mean of the post-event monthly NDVI values to generate $V_{post}$, then apply a similar procedure to the pixel-wise NDSI values to calculate the mean of the post-event monthly NDSI, $S_{post}$. This allows us to construct maps of the pixel-wise values of $dV$, $V_{post}$, $S_{post}$ and $P_t$ (Figure 1d) and thus to evaluate Equation 6. The full routine runs in GEE in less than 30 minutes for an area of ~$10^4$ km$^2$ (c. $10^7$ pixels).

**3.3 Performance testing**

We evaluate ALDI performance in terms of its ability to reproduce the location and size of manually-mapped landslides. For each earthquake inventory we define a study area based either on the area defined by the manual mappers (e.g., excluding areas where cloud or snow cover hampered manual mapping); or, where this is not available, on a convex hull that bounds the landslide inventory.

ALDI returns a continuous relative measure of the certainty with which a pixel is classified as a landslide. To evaluate this measure against a manually mapped landslide inventory it must be converted into a binary classification by thresholding the classification surface. The manual map is then rasterized to the same resolution as the classification surface - in this case, 30 m - using a 'majority area' rule, whereby landslide pixels are those with the majority of their area overlapped by landslide polygons. The benefit of a given classification can then be quantified in terms of success in identifying positive (landslide)

and negative (non-landslide) outcomes on a pixel-by-pixel basis. Thresholding the classification surface is a difficult exercise involving a trade-off between sensitivity, the fraction of the landslides that should be captured (also known as the true positive rate, TPR - the number of true positives normalised by all positive observations); and specificity, the number of false positives

that should be allowed in doing so (also known as the false positive rate, FPR - the number of false positives normalised by all negative observations). In practice, this threshold is often set by external requirements in terms of a desired sensitivity or

specificity, but these requirements can vary considerably between users and applications.

Receiver operating characteristic (ROC) curves provide a more complete quantification of the performance of the classifier (e.g., Frattini et al., 2010). The ROC curve is constructed by incrementally thresholding the classifier and evaluating true and false positive rates at different threshold values to generate a curve where the 1:1 line reflects the naïve (i.e. random) case. The true and false positive rates are insensitive to imbalanced data and thus are well suited to evaluation of landslide classification,

which typically has many more non-landslide than landslide pixels (García et al., 2010). The area under the curve (AUC) tends to 1 as the skill of the classifier improves towards perfect classification and to 0.5 as the classifier worsens towards the naïve (random) case. The strength of AUC is that it avoids the need to threshold the classifier and is widely used, enabling comparison with other landslide detection methods; its main weakness is that it is difficult to interpret in absolute terms. What AUC constitutes 'good' performance?

In our case, we seek to establish whether automated detection performance is such that it can be used as an alternative to manual mapping. However, it is difficult to compare the ALDI output against manual mapping because manual mapping is itself being used as the 'ground truth' in the absence of a better alternative. To address this, we first test the agreement between manual inventories in terms of true and false positive rates. $TPR_{I1-2}$ indicates the fraction of landslides in inventory I1 that are also predicted by I2 and $FPR_{I1-2}$ indicates the fraction of non-landslide pixels in I1 that are 'incorrectly' identified as landslide

pixels by I2.

ALDI performance in identifying landslide location on a pixel-by-pixel basis can then be compared against one of the manual maps as a competitor with the other manual map used as the check dataset. To enable the comparison, we first threshold the ALDI output to generate a binary classifier with the same FPR as the competitor inventory with respect to the check inventory. The ability of ALDI to successfully identify more landslide pixels than the competitor inventory can then be calculated from

the difference in their true positive rates, $TPR_{diff}$:

$$TPR_{diff} = TPR_{ALDI} - TPR_{Comp} \ , \ \ FPR_{ALDI} = FPR_{Comp} \tag{10}$$

where $TPR_{ALDI}$ and $FPR_{ALDI}$ are the ALDI true and false positive rates, respectively, both calculated from the check inventory; and $TPR_{Comp}$ and $FPR_{Comp}$ are the true and false positive rates for the competitor inventory, also calculated from the check inventory. The magnitude of $TPR_{diff}$ indicates the similarity in performance while the sign indicates the best performer (positive

values indicate that ALDI out-performs manual mapping and vice versa). This approach allows direct comparison between ALDI and manual mapping for the same classification threshold. Other metrics could be derived from the confusion matrix (e.g. Tharwat, 2020; Prakash et al., 2020) but these typically require assumptions about the relative weight assigned to true and false positives and negatives. Our approach avoids these assumptions because the ALDI output is thresholded to ensure that FPRs are equal to those of the competitor inventory.

In addition, we express spatial mapping error between manual inventories as the ratio of the intersection of the two maps to their union. This is equivalent to the 'degree of matching' (Carrara et al., 1992; Galli et al., 2008) and can be interpreted as the percentage of total mapped landslide area that the inventories have in common.

To examine the ability of ALDI to recover landslide size information we compare the area-frequency distributions of landslides from each manual map with those for landslides detected by ALDI. For manually-mapped inventories this information is

generally captured automatically since landslides are mapped as discrete objects rather than on a pixel-by-pixel basis. However, automated classifiers like ALDI require additional steps to convert a continuous pixel-based classification surface to a set of landslide objects. First, we generate a binary prediction of landslide presence or absence by thresholding the ALDI classification surface to match the manually-mapped FPR, as described above. The manual inventories examined here typically have very low FPRs (<2% of TPR on average and <7% at most, Table 3). Second, we convert the binary landslide map to a

set of landslide objects by identifying connected components at the 30 m resolution of the Landsat imagery (Haralick and

Shapiro, 1992). This connected components clustering is one of the simplest of many possible clustering algorithms. Finally, we calculate the area of individual landslide objects from the number of pixels in each object (cluster) and generate an area-frequency distribution.

## 3.4 Parameter calibration and uncertainty estimation

The ALDI landslide classifier has seven tuneable parameters: cloud threshold ($T_{cloud}$), pre-event stack length ($L_{pre}$), post-event stack length ($L_{post}$), snow threshold ($T_{snow}$), and the three exponents ($\alpha$, $\beta$ and $\lambda$) that control the weighting assigned to the $V_{post}$, $dV$ and $P_t$ layers, respectively. Calibrating the parameters and estimating the associated uncertainty is important because the parameters are difficult or impossible to set *a priori* and because we seek to develop a general model that can be applied to new landslide events not examined here. Our calibration seeks to optimize classifier performance evaluated by comparing the

classifier to 11 manually mapped landslide inventories using the performance metrics described in Section 3.3.

We calibrate ALDI parameters using one-at-a-time calibration for parameters that are internal to the GEE routine ($T_{cloud}$, $L_{pre}$, $L_{post}$), since these parameters are well constrained (in the case of $T_{cloud}$ and $L_{post}$) or have a limited number of possible values (in the case of $L_{pre}$ and $L_{post}$). We use an informal Bayesian calibration procedure (e.g., Beven and Binley 1992) for parameters in Equation 6 ($T_{snow}$ $\alpha$, $\beta$ and $\lambda$) since these parameters are less well constrained but evaluation of Equation 6 is computationally

cheap. We calibrate $L_{post}$, $L_{pre}$, and $T_{cloud}$, one-at-a-time (in that order) for each earthquake event then test alternative near-optimum parameter combinations to minimise the effect of the calibration order. These combinations are obtained by varying $L_{post}$ by +/- one year for optimum values of $L_{pre}$ and $T_{cloud}$, and doing the same for $L_{pre}$ at optimum values of $L_{post}$ and $T_{cloud}$. For each GEE run in the one-at-a-time process we run 500 simulations of Equation 6 with $T_{snow}$ and $\alpha$ randomly sampled from uniform probability distributions and the ratio parameters sampled from uniform distributions of log10($\alpha$:$\beta$) and log10($\alpha$:$\lambda$).

We sample the ratio parameters in logarithmic space to maintain symmetric sampling density with distance from a ratio of unity (e.g. $\alpha$:$\beta$=0.1 where $\beta$=10$\alpha$ should be sampled as densely as $\alpha$:$\beta$=10 where $\alpha$=10$\beta$).

We examine $L_{post}$ of up to five years because vegetation typically begins to re-grow over this timescale (Restrepo et al., 2009), and $L_{pre}$ of up to ten years because we expect that other landscape changes (e.g. fire, drought and landslides caused by other triggers) will begin to disrupt the pre-event signal at longer timescales. In both cases we examine only integer year values to

ensure consistent sampling within the monthly bins. We use the full range of NDSI values for $T_{snow}$ ([0,1]) and cloudscore values for $T_{cloud}$ ([0,1]). For the three exponents, we use zero for the lower bound and iteratively refine the upper bound to ensure that optimum performance at any site is found to be within the range.

We perform the calibration for individual earthquakes to estimate the optimum classification skill that could be obtained when calibrating on all the check data. We then retain the best 20 parameter sets (measured in terms of AUC) from each earthquake

to generate a global set of 100 parameter sets. To account for parameter interaction (particularly between the three exponents $\alpha$, $\beta$ and $\lambda$) within a set we retain parameter sets as 7-element vectors. To ensure that each manually-mapped landslide inventory is given equal weight as a check dataset we calibrate to each in turn taking 7 parameter sets from calibration to each of the three Gorkha inventories, and 10 from each of the two inventories at the other sites. Finally, we run ALDI with each of these 100 parameter sets to generate 100 ALDI classification surfaces then take the mean for each cell.

To simulate 'blind' application of ALDI to future events, we perform a holdback test in which we run ALDI using the global parameter set but holding back the 20 parameter sets that were derived from the site at which testing is being performed. In this test the parameters used to run ALDI are un-influenced by the specific behaviour of the test site.

## 4 Results

### 4.1 Spatial agreement: Gorkha case study

We first illustrate our approach using the 2015 Gorkha earthquake, where three manual inventories are available, and then consider the other four earthquakes introduced in Section 2. All three manual inventories for the Gorkha earthquake show an elongated cluster of landslides extending from northwest to southeast (Figure 2a) that coincides with the area of steep slopes that experienced the most intense shaking. However, when the maps are compared at a finer scale they differ considerably (Figure 2c,e). In some cases, one mapper has identified a landslide but one or both of the others have not (e.g., location A in

Figure 2e). Some, but not all, of these missed landslides can be attributed to areas where imagery was unavailable or where the ground was obscured by cloud (shown as grey areas in Figure 2c). In other cases, mapped landslides overlap but their size and/or shape differ, due either to differences in interpretation of landslide boundaries (e.g., location B in Figure 2e) or to the georeferencing of the underlying imagery from which the landslides were mapped. Georeferencing differences seem particularly likely to explain mapped landslides of very similar size and shape that are offset by small distances (e.g., location

C in Figure 2e), or appear distorted relative to one another so that their outlines only partially overlap (e.g., location D in Figure 2e).

The ALDI classifier applied to the Gorkha earthquake captures the broad spatial pattern of mapped co-seismic landslides with large patches of high ALDI values, and thus high classification likelihood, corresponding to clusters of mapped landslides (Figure 2b). Examining a subsection of the study area (Figure 2d) shows that ALDI identifies the same broad zones of more

intense landsliding as identified in the manual mapping. The ALDI output also contains a series of stripes ~1 km apart and ~150 m wide trending west-northwest to east-southeast, however, and most clearly visible across the centre of the map. These are the result of data gaps in Landsat 7 images since 2003 due to Scan Line Corrector (SLC) failure on the Landsat 7 sensor. Although both pre- and post-event image stacks include Landsat 5 and 8 images in addition to Landsat 7, these data gaps clearly influence the ALDI output, with high values more likely for pixels where Landsat 7 data are not available.

Zooming in to a smaller subsection of the study area suggests that most of the landslides that are included in both inventories overlap areas of high ALDI values (Figure 2e). In addition, areas of high ALDI values overlap many of those landslides identified by one inventory but not the other, although there are mapped landslides that do not overlap areas with high ALDI values (Figure 2e). In many cases, the patches of high ALDI values have shapes that closely follow those of the mapped landslides (Figure 2e). In other cases, patches of high ALDI values have typical landslide morphology but are not in either

inventory (e.g., location E in Figure 2e), raising the question of whether these should be considered genuine classifier false positives or are in fact landslides missed in all three manual maps. Given that each inventory misses landslides identified by another, this possibility cannot be excluded. In other cases, the patches of high ALDI values have a size and/or shape that suggests that they are misclassifications. These may be due to cloud, shadow, snow or other landscape changes not associated with landslides (e.g., crop harvesting, river channel change, building construction).

### 4.2 ALDI calibration: Gorkha case study


In this section, we seek to establish the best possible ALDI performance when parameters can be optimised to a single study site and identify the influence of parameters on that performance, both in terms of sensitivity to the parameter and preferred range for the parameter. We illustrate this using the Gorkha earthquake, calibrating ALDI's seven tuneable parameters (columns A-G in Figure 3) to optimise agreement with two of the manually mapped landslide inventories measured using our

two performance metrics (rows in Figure 3). The results are visualised in Figure 3 using dotty plots (after Beven and Binley, 1992): a matrix of scatter plots where each subplot shows model performance (y-axis) against a parameter value (x-axis). The histogram above each scatter plot shows the frequency distribution of parameter values for the best 50 model runs for that metric and check dataset.

All the scatter plots in Figure 3 show wide scatter in performance for a single value of any given parameter, indicating that the model is sensitive to multiple parameters. However, the key feature of each plot is the upper bound on ALDI performance for a given parameter value, and its sensitivity to change in that parameter. This upper bound can be interpreted as the best possible ALDI performance at value x of parameter A when all other parameters are given flexibility to optimise. Plots where this upper bound is near horizontal suggest limited influence of a particular parameter and are accompanied by broad histograms. Narrow peaks in a plot's upper bound indicate that good model performance requires that parameter to be set within a narrow range, with performance degrading rapidly as values depart from this range independent of other parameter values. In the following paragraphs we examine the influence of each parameter in turn (Figure 3).

Setting the pre- and post-earthquake stack lengths ($L_{pre}$ and $L_{post}$ respectively) involves a trade-off between: errors caused by landslides (or other landscape changes) not associated with the earthquake, if the stack is too long; and errors caused by cloud cover, if the stack is too short. For the Gorkha earthquake, ALDI performance is most sensitive to $L_{post}$, indicated by the steep gradient in upper bound performance across both metrics and for all check datasets (Figure 3, column G). For all metrics and datasets, a post-earthquake stack length of only one year produces the best performance. This may be because longer stacks are more likely to include other landscape changes after the earthquake that disrupt the signal, such as post-seismic landslides or re-vegetation of co-seismic landslides.

ALDI allows landslides to be identified only in pixels where NDSI is lower than the snow threshold ($T_{snow}$). ALDI performs well (i.e. <20% from optimum) for $T_{snow}$ values ranging from 0.1 to 0.9 (Figure 4, column D). For TPR$_{diff}$ the best values of $T_{snow}$ are 0.2-0.4 with a rapid decline in performance as $T_{snow}$ is reduced and a slow decline as it is increased (Figure 4, panels D1-2 and D3-4). This suggests that snow rarely causes false positives even when little effort is made to remove it, but that an overly conservative snow threshold results in landslides being misclassified as snow. The AUC metric behaves similarly to TPR$_{diff}$ with a larger performance reduction at low $T_{snow}$ values and reduced performance reduction at high $T_{snow}$ values (Figure 4, D5-6).

The α:β ratio controls the influence of change in NDVI ($dV$) relative to mean post-earthquake NDVI ($V_{post}$). Noting that $dV$, and $V_{post}$ are bounded to be <1, and that by definition β=α(α:β)$^{-1}$, larger values of α:β result in smaller exponents on $V_{post}$ and larger values of the term. ALDI is thus dominated by $V_{post}$ at higher α:β ratios, and by $dV$ at lower ratios. There is a clear optimum within the parameter space and a large reduction in performance away from this optimum indicating that both layers ($dV$ and $V_{post}$) are important components of the classifier (Figure 3, column B). Best performances are found in the range α:β = 3-4 for TPR$_{diff}$ and in the range α:β = 10-20 for AUC, suggesting that more weight needs to be given to $V_{post}$ to successfully identify landslides, particularly when bulk performance over the full ROC curve is of primary concern.

The α:λ ratio controls the influence of change in NDVI ($dV$) relative to the likelihood that the $dV$ values in the post-event stack are significantly different from the pre-event stack ($P_t$). As explained above, ALDI is dominated by $P_t$ at higher α:λ ratios, and by $dV$ at lower ratios. ALDI performance is somewhat sensitive to this parameter for both TPR$_{diff}$, and AUC, with gentle but consistent slopes to the upper bound performances (Figure 3, column C). Best performances are found for α:λ in the range 0.01-1 for TPR$_{diff}$ and 0.1-5 for AUC, suggesting that, although both layers contribute important information, $dV$ is a stronger predictor than $P_t$ for the Gorkha case study.

Optimum parameters for the Gorkha study site differ slightly between performance metrics (compare histograms down columns in Figure 3). This reflects the different focus of the metrics, where TPR$_{diff}$ gives the strongest weight to very conservative (i.e., low FPR) classification thresholds (Figure 3, rows 1-2), and AUC weights all classification thresholds equally (Figure 3, rows 5-6). In general, the parameters to which ALDI performance is most sensitive are also those for which optimum values are most robust to changes in check dataset or performance metric. For example, there is negligible change in optimum values for $L_{post}$ and $T_{snow}$ across the range of metrics and datasets. α:β and α:λ are both broadly comparable between metrics although in both cases there is a shift towards higher optimum values for AUC, indicating that for this metric NDVI

difference is less important than it was for TPR$_{diff}$ (noting that the improvement is always <3%). α:β has a progressively less clear optimum as metrics become more generalised (from TPR$_{diff}$ to AUC) indicating reduced parameter sensitivity for AUC. $T_{cloud}$ and $L_{pre}$ have larger changes in optimised parameters between the metrics, although the sensitivity to these changes is small in performance terms (Figure 3, columns 5-6). Optimum $T_{cloud}$ is 0.7 for TPR$_{diff}$ but 0.5 for AUC, optimum $L_{pre}$ is in the range 2-5 for TPR$_{diff}$ and 5-10 for AUC. ALDI performance is insensitive to α, varying by <10% across the parameter range for all metrics, generating a broad histogram of best-performing parameter values and showing large shifts in optimum value depending on both the metric and the dataset used to assess performance (Figure 3, column A).

### 4.3 ALDI calibration: global comparison

We focus our global comparison on the AUC performance metric. Results for TPR$_{diff}$ are very similar and can be found in the supplementary information (Figures S1-S6). Figure 4 shows that optimum values for a given parameter differ between sites; that sensitive parameters at one site are usually sensitive at others; and that absolute performance differences between different inventories at a site can be large, although the trends are generally similar.

ALDI is sensitive to $L_{post}$ for all sites but with trends that differ between sites: for Haiti and Gorkha one year is best, two years is reasonable and three years is poor; for Kashmir and Wenchuan one year is best but two also gives reasonable results; for Aisen five years is best and one year is particularly poor (Figure 4 column G). An $L_{post}$ of two years generally results in fairly good performances for all five sites. These site-by-site differences suggest a connection between the optimum time series length $L_{post}$, the frequency of Landsat image acquisition during the study period, and the processes that cause NDVI change at different sites (e.g., vegetation growth rates, fire, drought or post-seismic landsliding). While this does not preclude good performance of ALDI using a global parameter set, it does imply that performance with this global parameter set will almost always be sub-optimal relative to a locally-calibrated set. However, such local calibration requires independent landslide mapping over at least part of the study area. Further work might seek to connect optimum parameters at a site with its image and landscape characteristics enabling a refinement of the parameters without the need for additional mapping.

ALDI is sensitive to $T_{snow}$ in three of the five sites and particularly for Aisen, but in all cases $T_{snow}$ of 0.5-0.8 results in performances that are at least close to optimum (Figure 4, column D). ALDI is only weakly sensitive to $L_{pre}$ for all sites and with subtly differing trends: for Kashmir three years is best, for Wenchuan and Haiti 10 years is best and for Aisen and Gorkha best performances are in the range of five to 10 years (Figure 4, column F). However, the trends are not linear and an $L_{pre}$ of five years generally results in fairly good performances for all five sites. ALDI is generally insensitive to $T_{cloud}$ across the range 0.3-0.7 with best performances consistently found at 0.5, although these are at most 10% better than those for other values in the range (Figure 4, column E). ALDI is insensitive to α alone, but is strongly sensitive to α:β and weakly sensitive to α:λ at all sites (Figure 4, columns A-C) with best performances found for α:β in the range 1-100.

ALDI application would be both faster and simpler if single optimum values could be used for the three pre-processing parameters within Google Earth Engine ($T_{cloud}$, $L_{pre}$, $L_{post}$). In particular, the shorter the post-event window $L_{post}$ the sooner an inventory can be compiled following an earthquake. Our site-by-site calibration suggests that it is possible to find single values for these parameters that result in good performance for all study sites (Figure 4). This is the case when the cloud threshold $T_{cloud}$ is 0.5, the pre-earthquake stack length $L_{pre}$ is 5 years, and the post-earthquake stack length $L_{post}$ is 2 years (thus it is reasonable to expect that an ALDI-derived inventory can be generated after 2 years). We also examined performance when these parameters were allowed to vary but found that the performance improvement for the global parameter set was negligible. To examine similarity between locally optimised parameters and compare them to a global set of parameter sets, we first identified the best 100 parameter sets for each study site, using AUC as the performance metric (Figure 5). To generate the global parameter sets we held $T_{cloud}$, $L_{pre}$ and $L_{post}$ constant at 0.5, 5 years and 2 years respectively; then, treating the remaining parameter sets as 4-element vectors, we sampled the best 20 parameters from each site; finally, we generated a holdback parameter set for each site by removing that site's parameters from the global set. Locally optimised parameter sets (grey

histograms in Figure 5) are broadly consistent with the global set (blue histograms) with a small number of exceptions: $T_{snow}$ should be set lower for Kashmir and higher for Aisen, α:β should be set higher for Kashmir and α:λ set lower for Gorkha. These differences are accentuated in the holdback distributions (the black outlined histograms) because the divergent local parameter values are stripped from the set, pulling the distributions away from their local optima. We would expect larger performance degradation from local to global to holdback parameter sets at sites where these distributions are more different. ALDI with locally optimised parameters always out-performs the global parameters and the global parameters always out-perform the holdback parameters (Table 3). The difference between local and global parameters is generally larger than between global and holdback parameters. In fact, performance reduction from global to holdback parameters is always <1% for AUC. This indicates that the five study sites provide an adequately varied calibration set to enable generation of a general parameter set that is not overly influenced by any one site. This is encouraging for future 'blind' ALDI application. However, the difference in performance between local and global parameters shows that local optimisation can improve ALDI performance in terms of AUC by up to 9% (and by 2% on average). In three cases, one for Kashmir and two for Gorkha, local optimisation improves ALDI to the point where it is no longer out-performed by the manually mapped competitor inventory but instead out-performs it in terms of identifying landslide locations in the check inventory. This is somewhat consistent with the observed divergence of locally optimised parameter distributions from the global distribution at these sites (Figure 5). However, it likely also reflects the broadly similar performance (i.e. skill) of ALDI and manual mapping at the sites (Table 3).

**4.4 Spatial agreement: global comparison to manual mapping**

Spatial agreement between manual landslide inventories is surprisingly low not only for the Gorkha study site shown in Figure 2 but across all sites. TPRs range from 0.08-0.8 indicating that at best 80% and at worst 8% of the landslide area mapped by one inventory is also identified as a landslide by a second test inventory (Figure 6a and Table 3). FPRs range from 0.0003-0.03, indicating that at best 0.03% and at worst 3% of the area that is identified as non-landslide in one inventory is instead identified as a landslide by a second test inventory. There are two possible reasons why FPRs are so much lower than TPRs: 1) landslide density is low, so there are few positives (TP+FN) and many negatives (TN+FP); these are the denominators of TPR and FPR, respectively, amplifying TPR and damping FPR; and 2) landslide mappers may be inherently conservative, mapping only features that they are confident are landslides. TPRs and FPRs are positively correlated but with considerable scatter (Figure 6a). In some cases manual maps agree quite closely: for example, the inventories of Gorum et al. (2013) and Harp et al. (2016) for Haiti ($H_{GH,}$ $H_{HG}$) or those of Zhang et al. (2016) and Watt (2016) for Gorkha ($G_{ZW}$, $G_{WZ}$). These cases have a relatively high TPR given their FPR and plot towards the top left of the point cloud in ROC space (Figure 6a). In other cases the agreement is weaker, such as between the inventories of Li et al. (2014) and Xu et al. (2014) for Wenchuan ($W_{LX}$, $W_{XL}$) or those of Sato et al. (2007) and Basharat et al. (2016) for Kashmir ($K_{SB}$, $K_{BS}$). There is a symmetry to the inventory comparison because each inventory takes a turn as the competitor dataset (to which ALDI is being compared) and as the check dataset (against which both are evaluated). As a result, a single pairwise comparison results in two points in Figure 6a reflecting the switching of roles. The three-way comparison for the Gorkha earthquake results in three pairwise comparisons and six points. When one inventory is considerably more complete and less conservative then the separation between pairs of points will be large (e.g., Watt and Zhang for Gorkha). Zhang et al. (2017) reported, in their metadata, that their inventory is incomplete and focusses on the largest landslides, while that of Watt (2016) was more complete and less conservative. As a result Zhang et al. (2016) successfully identified only 10% of the landslide pixels identified by Watt (2016) but identified only a tiny fraction (<0.1%) of the study area as landslides when Watt (2016) considered that they were not ($G_{ZW}$ in Figure 6a). Conversely, Watt (2016) successfully identified 80% of the landslides identified by Zhang et al. (2016), but also identified a further 1% of the study area as landslides that were not identified as such by Zhang et al. (2016) ($G_{WZ}$ in Figure 6a).

To evaluate ALDI performance relative to manual mapping, we compare the ability of ALDI to successfully identify more landslide pixels in one (check) inventory than another (competitor) inventory when ALDI output is thresholded to reproduce

the FPR of the competitor inventory. This TPR difference (TPR$_{diff}$) is shown as a red line in Figure 6b-f; positive differences indicate that ALDI out-performs manual mapping and vice versa. ALDI out-performs manual mapping in the majority of cases when parameters are locally optimised (10 of 14 cases, Figure 6 and Table 3) and is comparable to manual mapping when a single global parameter set is applied to all study sites (8 of 14 cases). Performance is only slightly reduced when the test site is held back from the global optimisation and ALDI continues to outperform manual mapping in 8 of 14 cases.

ALDI performs better at some sites than others, with performances for Aisen and Gorkha particularly good (Table 3). Performance is poor for Haiti, both in absolute terms and relative to the manual mapping. For AUC, an indicator of absolute performance, ALDI performance for the Haiti case is ranked 10th-11th of 14 (where the range results from combining local global or holdback tests). Relative to manual mapping, ALDI correctly identifies 51-74% fewer landslide pixels for the same FPR. Explanations for these performance differences are discussed in Section 5.4. ALDI in Wenchuan performs only

moderately in absolute terms, with ranked performances in the range 9th to 12th out of 14 for AUC, but out-performs manual mapping (1st and 4th for TPR$_{diff}$) as a result of the relatively poor agreement between manual maps for the site. Kashmir has very marked differences in ALDI performance depending on the test dataset (all <4th of 14 for Sato et al. (2007); all >9th of 14 for Basharat et al. (2016)), illustrating the difficulty of interpreting performance relative to check data when the check data themselves contain errors of similar magnitude to the data being tested.

**4.5 Area-frequency distributions**

Probability density functions for manually-mapped landslide areas (Figure 7a-e) follow a consistent distribution with a roll-over and a heavy right tail that is approximately linear in logarithmic space but that usually has positive (convex up) curvature or a roll-off at very large areas. These characteristics have already been widely reported both for the study inventories in particular (e.g., Gorum et al., 2013; Li et al., 2014; Roback et al., 2018) and for many other landslide inventories worldwide

(e.g., Tanyas et al., 2019). Different inventories for the same study site show broadly consistent scaling in their right tail but tend to differ markedly in the location of the roll-over, modal size, degree of curvature in their right tail and the location (and presence) of a roll-off for very large areas (e.g., Figure 7a, d and e). These differences, and their possible explanations, have also been widely reported for these and other sites (see review by Tanyas et al., 2019).

The area-frequency distributions derived from ALDI reflect the sizes of clustered landslide-affected areas (rather than the areas

of landslide objects themselves). The ALDI-based distributions generally exhibit a broadly similar right tail to those of the manually-mapped distributions; both have heavy right tails that closely approximate a power law, and have similar scaling (i.e. slope in logarithmic space) in that right tail. However, the ALDI-based distributions, are clearly different from those derived from manual mapping, they lack: 1) the roll-over at small areas (in all cases, Figure 7a-e); 2) the positive curvature to the right tail (particularly clear for Haiti, Figure 7d); and 3) the roll-off at very large areas (resulting in oversampling of

landslides >$10^5$ m$^2$ for Wenchuan, Figure 7c).

These differences can be explained in terms of amalgamation and censoring. Amalgamation of multiple neighbouring landslides increases the frequency of large landslides, fattening the right tail (Marc and Hovius, 2015); and in some cases considerably increasing the size of the largest landslide (e.g. Aisen and Wenchuan, Figure 7b-c). Re-sampling to a 30 m grid makes it impossible to record landslides smaller than a single pixel (i.e. 900 m$^2$), censoring them from the area-frequency

distribution.

To illustrate the role of amalgamation and censoring we convert the manual landslide maps to binary grids at 30 m resolution, using a 'majority area' rule to identify landslide-affected pixels, and perform the same connected component clustering used for ALDI. Resampling to 30 m should result in strong censoring and some amalgamation as explained above. Re-clustering with a connected components algorithm likely results in further amalgamation. Figure 7 shows that resampling and re-

clustering manually-mapped landslides transforms their area-frequency distributions removing the rollover and resulting in distributions that are very similar to those for landslide pixels classified with ALDI. This supports our interpretation that misfit

between ALDI and manual mapping is due to censoring and amalgamation, although we are unable to determine their relative roles. Misfits due to the resolution of the Landsat and thus the classification surface are difficult to overcome whereas improvements in clustering could be more easily implemented.

## 5 Discussion

### 5.1 The problem of testing landslide location against uncertain check data

The $TPR_{diff}$ results for the five study sites show that ALDI out-performs manual mapping in 8 of 14 inventories in terms of its ability to identify landslide-affected areas identified in a second check inventory. This may indicate that ALDI is more skilful than each of these inventories at identifying the locations of landslides. However, because the check inventories are themselves known to contain error, this is not a secure result; erroneous out-performance by ALDI would result if it identified the same artefacts that had been (erroneously) mapped in the check dataset but not in the competitor.

A more secure result can be obtained from the four (of seven) inventory pairs where ALDI out-performs both inventories in the pair when the other is used as check data. This indicates that the ALDI output is more similar to each inventory than the inventories are to one another (Table 3) and demonstrates that ALDI must be more skilful than at least one of the inventories (either the check or competitor inventory) in identifying the locations of landslides. However, we are still unable to conclude whether ALDI is better than one or both inventories, or identify which inventory is better. This is because errors in a single inventory influence the result both when it is used as the predictor (i.e., as a competitor against ALDI) and the check dataset (against which both are evaluated).

### 5.2 Spatial disagreement in manually mapped inventories reflects processing errors, not solely mapping errors

Our findings on the large locational mismatch between co-seismic landslide inventories are initially surprising, given the widespread assumption that such inventories represent a 'ground truth' and the limited attempts to propagate these errors into hazard maps, classification tests, process inferences, or landslide rate estimates. However, the limited number of other studies that do quantify landslide inventory error all suggest very weak spatial agreement between landslide inventories (Ardizzone et al., 2002; Galli et al., 2008; Fan et al., 2019).

The process of generating a landslide inventory from satellite imagery involves choosing which images to map from and how to post-process and georeference them before landslides can be identified and delineated by a human mapper. Thus, the comparison of two inventories is not a direct test of the consistency with which human mappers detect and delineate landslides but instead the consistency with which different research groups generate landslide inventory maps. As an illustration of this distinction, Fan et al. (2019) found that landslide inventories had an overlap of 67%-86% (and 76% on average) when comparing between mappers in the same team mapping from the same imagery. This differs considerably from both our own results (8-30% overlap, Table 3) and other published cross-inventory comparisons (19-44% overlap, Ardizzone et al., 2002; Galli et al., 2008; Fan et al., 2019). In these cases, the inventories being compared were published by independent research groups and were not only collected by different mappers without collaboration but were generated from different sets of satellite images. For example, Roback et al. (2018) used Worldview imagery with high spatial resolution but which suffer from severe distortions in the Gorkha study area due to the steep landscape and oblique look angles (Williams et al., 2018). Even if landslides were correctly identified in both sets of imagery, differences between inventories could be introduced during georeferencing. Figure 8 shows evidence of the same problem for the Wenchuan inventories, where two sets of mapped landslides with strikingly similar patterns are offset by ~1 km. These georeferencing errors are difficult to attribute to a single inventory and appear to vary in magnitude and direction even over quite short length scales within an inventory (Figures 2 and 8). Thus, improved performance of ALDI relative to a particular inventory reflects an improved overall workflow rather than specifically the ability to identify landslides in images.

## 5.3 Limitations to ALDI performance

ALDI performance varies from site to site, with particularly good performances for Aisen and Gorkha, but particularly poor for Haiti. The overall poor performance for Haiti may reflect the drier conditions in the study area, which lead to vegetation that is more difficult to differentiate from landslide scars, or the higher degree of human influence on land cover relative to other sites, which may result in more vegetation changes not related to landslides. ALDI can identify landslides only in areas where they result in a change in NDVI and will perform better in areas where this change is more pronounced (all else being equal). This will occur where pre-event NDVI is higher due to denser and/or more vigorous vegetation coverage, both of which result in a larger share of reflectance from leaves, with their more pronounced 'red edge' (the red to near infra-red reflectance change). Conversely, ALDI will perform poorly in areas with sparse vegetation such as the epicentral area of the 2010 Sierra Cucapah earthquake (Barlow et al., 2015).

Poor performance for Haiti in comparison with the manual mapping may also be due to ALDI's coarse 30 m resolution relative to the dimensions of the landslides in the study area. ALDI will identify a pixel as landslide affected only if the landslide occupies enough of the pixel to alter its spectral response, and will perform better when landslides are large enough to occupy large fractions of one or many pixels. Given their typically elongate shape (Taylor et al., 2018), landslides with widths <30 m and thus areas <2,700 m$^2$ (assuming L/W=3, 75$^{th}$ percentile from Taylor et al., 2018) will be partially censored, with the degree of censoring increasing as width declines. Median landslide area in the inventories examined here ranges from 250 m$^2$ for Haiti (Harp et al., 2016) to 19,000 m$^2$ for Kashmir (Basharat et al., 2016), with medians less than 2,700 m$^2$ in 4 of 14 inventories. Therefore, this censoring will strongly affect ALDI-derived inventories, particularly in areas with lower relief (such as Haiti) where smaller landslides are expected to be more common (Jeandet et al., 2019).

Finally, poor performance for Haiti is also likely to reflect the limited number and quality of Landsat images acquired over the study area. ALDI used imagery from 2005-12 to identify landslides triggered by the Haiti earthquake and thus relies exclusively on Landsat 5 and 7 data (Landsat 8 launched in 2013). Both Landsat 5 and 7 are problematic for this study site and period. All of the Landsat 7 data contain data gaps due to Scan Line Corrector (SLC) failure from June 2003 onwards and only small amounts of Landsat 5 data for areas outside the USA were retained during this period, limiting archival imagery in some areas (see Figure S5 in Pekel et al., 2015). For Haiti the pre-earthquake stack is composed of 6 Landsat 5 images and 205 Landsat 7 images and the post-earthquake stack of 16 and 91 images, respectively. Limited availability of Landsat 5 data at this site means that in some areas the classifier relies exclusively on Landsat 7 and is thus unable to calculate an ALDI value for pixels within the data gaps (these are visible as white stripes in the eastern half of Figure 9b). While some areas of high ALDI values show good agreement with mapped landslides, there are also large patches of high ALDI values with complex shapes that are uncharacteristic of landslides and that manual mapping shows as likely false positives (Figure 9c).

Given these limitations to Landsat 5 and 7 imagery, it is perhaps surprising that ALDI performs so well in the Aisen case (where the stack extends from 2002-2009). This is likely due to the larger number of Landsat 5 images available for the study site (140 in the pre-earthquake stack and 46 in the post-earthquake stack) and to the location of the area of densest landsliding near the centre of a Landsat 7 image where data gaps related to SLC failure are minimised. The 2015 Gorkha earthquake is the only case study for which Landsat 8 data were available, perhaps explaining the relatively good performance at this site and offering hope for application to more recent events.

Sparse image data (associated with incomplete archiving of Landsat 5) and sensor problems (primarily SLC failure on Landsat 7) from 2003-2014 suggest ALDI-based mapping in this period should be handled with care. However, the majority of our test earthquakes come from this period and we have demonstrated that even with these constraints, ALDI performs well in determining landslide locations for four of the five case studies, both in absolute terms and relative to manual mapping. Potential checks on ALDI applications during this time period could entail careful checking of the numbers of images in the pre- and post-earthquake stacks, the extent of Landsat 7-derived striping in the ALDI map, and the size and shape of the landslides in the ALDI-derived inventory. Small image stacks (particularly for Landsat 5), extensive striping, and large

complex landslide shapes should all be treated as indicators of potentially poor ALDI performance. However, even when large image stacks are available for an earthquake-affected area, cloud cover can limit the number of usable observations per pixel within the pre- and post-earthquake stacks.

ALDI can identify landslide-affected pixels with a high degree of skill (comparable to manual mapping) but is considerably less skilful in identifying discrete landslides, as demonstrated by the difference in ALDI and manually-mapped area-frequency

distributions. As with Parker et al. (2011), additional steps are required to identify separate landslides (e.g., Marc et al., 2016). Calibration based on a small subset of manually-mapped landslides followed by subsequent manual editing to remove false positives could result in a very good inventory in a fraction of the time associated with full manual mapping.

**5.4 Strengths and weaknesses of ALDI relative to manual mapping**

The most widely used properties of landslide inventories are landslide location and geometry (Guzzetti et al., 2012). In terms

of location, ALDI performs comparably to manual mapping in identifying whether the majority of each pixel in a 30 m grid is landslide-affected. However, it performs worse in capturing landslide area-frequency distributions, primarily because it cannot identify small isolated landslides (i.e. with areas <900 m$^2$ separated by more than 30 m) and separating the output from ALDI (or any other pixel-based classifier) into discrete landslide objects is not straightforward.

Current approaches to train and test landslide prediction models (including hazard and susceptibility models) almost

exclusively use pixel-based information on landslide presence or absence rather than information about the size or shape of a landslide at a particular location (see Bellugi et al. (2015) for an exception). For such applications, skilful identification of landslide-affected pixels is the sole requirement. Our results suggest that the ALDI landslide inventory would be an appropriate product to use in these cases as it is better than at least one of the manual inventories in four of the five case studies (Table 3). Landslide geometry is required to construct landslide are-frequency distributions, and is useful to distinguish landslide

initiation and runout zones (Marc et al., 2018). Manual mapping provides landslide geometry with a high level of accuracy, although disagreements in landslide area-frequency distributions for manually-mapped inventories have already been reported, with most pronounced differences being in roll-over location, usually due to differences in image resolution (Galli et al., 2008; Fan et al., 2019; Tanyas et al., 2019). The accuracy of landslide geometry derived from ALDI depends strongly on the extent to which landslide pixels can be clustered to identify separate landslides (e.g., Marc et al., 2016) and on the pixel resolution.

The first of these is common to all pixel-based classifiers. Given the relatively coarse resolution of the underlying Landsat data, we expect ALDI-derived geometries to be accurate only for large landslides, as shown in Figure 7.

All in all, we expect ALDI to be useful in identifying areas for further (more detailed) mapping at multiple scales: 1) globally, as a supplement to the existing archive of co-seismic landslide inventories by examining historic events for which a landslide inventory has never previously been generated but where landslides are known or expected to have been triggered; 2) at a site,

to identify areas of interest or to extend the study area beyond that which can be feasibly mapped by hand; 3) at the finest scale, to identify individual candidate landslides to be manually checked and re-digitised if necessary. We also expect ALDI to be a useful check on manual mapping, enabling increased homogeneity in areas where there is only patchy coverage of high resolution imagery and perhaps identifying georeferencing errors.

We do not expect ALDI in its current form to be as useful as manual mapping: 1) as a source of rapid landslide information to

inform emergency response (because ALDI performs better with two years of post-event images); 2) for size or shape distributions (because of censoring and amalgamation inherent in 30 m pixel-based output); 3) for analysis where landslide initiation zones must be differentiated from runout; 4) in landscapes where vegetation is sparse (because NDVI changes in landslide pixels are unlikely to be detectable relative to natural variability); and 5) in landscapes where small landslides are widely distributed across the landscape (because the pixel-averaged NDVI change will be small if only a fraction of a pixel is

disrupted).

## 5.5 Comparison to other automated detection methods

Automated detection of landslides typically relies on vegetation change detection and involves either generating indices of surface disturbance from which landslides can be manually identified (e.g. Scheip and Wegmann, 2020), or performing a supervised classification (e.g. Barlow et al., 2003; Behling et al., 2014; 2016; Prakash et al., 2020).

A recent example of automated surface disturbance detection, HazMapper (Scheip and Wegmann, 2020), uses similar image data (Landsat) and the same platform (Google Earth Engine) as ALDI, but for a different purpose and using different functions to combine and transform the imagery. HazMapper is designed to generate a qualitative metric for surface change rather than a landslide-specific mapping tool. As a result, the approach does not mask snow-covered areas in case these are of interest for a user's particular application. The approach is simpler than that of ALDI in that HazMapper calculates the NDVI difference

only, rather than accounting for post-event NDVI, seasonal variability and noise in the NDVI signal for each pixel. It is currently only applied to Landsat 7 onwards and only for individual sensors, rather than combining images from multiple Landsat sensors. This limits the events that can be examined to those occurring after 1999. However, results from HazMapper for the same study periods examined here show a good qualitative agreement with the ALDI results. The similarity in approach, using stacks of Landsat imagery before and after a suspected trigger event, means that the two approaches will likely have

many of the same strengths (e.g., the accurate georeferencing of Landsat imagery) and limitations (e.g., the coarse resolution of Landsat imagery and long wait times required to generate the post-event stack).

Alternative approaches to landslide detection that involve supervised classification typically rely on machine learning (e.g. Prakash et al., 2020) or clustering methods (e.g. Barlow et al., 2003; Behling et al., 2014; 2016). These more complex approaches are compatible with the data and platforms that we use here. Although we have taken a simpler approach, the

classification surfaces generated by ALDI could be coupled with modern machine learning approaches to improve ALDI's landslide detection skill. However, our results also highlight an important potential limitation to the use of supervised learning for landslide detection in general. Given the very severe disagreement between manually-mapped landslide inventories, any supervised learning method will have a very high risk of propagating gross errors into the classifier unless the training inventory is precisely co-located with the imagery used by the classifier. ALDI could help improve existing supervised classification

efforts by providing additional well-referenced landslide inventories, or by correcting existing ones.

## 5.6 Application to future earthquakes

Increased frequency and quality of optical imagery suggests that ALDI should perform well for future earthquakes. In particular, Sentinel 2 imagery can generate NDVI at 10 m spatial resolution (Table 1). The two Sentinel 2 satellites were launched between June 2015 and March 2017, and thus there is a limited stack of pre- or post-earthquake images available to

date. The 2018 Hokkaido earthquake offers the best trade-off to date between pre- and post-event data. As a test of the wider applicability of ALDI to future events, we ran ALDI using the global parameter set identified above, and evaluated its results against landslides mapped from aerial imagery by Wang et al. (2019). The results are extremely promising both at the scale of the entire epicentral area (Figure 9d and e), and of individual landslides, with few false positives, a large area under the ROC curve (0.94), and many landslides clearly delineated by a sharp break from high to low ALDI values (Figure 9f).

## 6 Conclusion

Rapid derivation of landslide inventories after large triggering events remains a key research challenge. We have developed a parsimonious automatic landslide classifier, ALDI, that uses pre- and post-event stacks of freely-available medium-resolution satellite imagery and relies on landslide-induced changes to vegetation cover and thus to NDVI values. We test the classifier against multiple independent manually-mapped inventories from five recent earthquakes. Considering that manually-mapped

inventories are typically assumed to be the 'ground truth' against which automatic classifiers are evaluated, we find that

agreement between different manual inventories is surprisingly low (8-30% of landslide area in common). ALDI often identifies landslides in one inventory missed in the other and even identifies some candidate landslides not in either inventory but that have location and morphometric characteristics that strongly suggest they are true positives.

We further find that ALDI can identify landslide locations with a level of skill that is comparable to manual mapping on a pixel-by-pixel basis. ALDI calibrated to mapped landslides at a site out-performs manual mapping in 10 of 14 cases (i.e. 71%). The only cases where manual mapping performs better are: the two inventories for the 2010 Haiti earthquake, where the stack of available Landsat images is extremely limited; and the cross comparison of inventories for the 2015 Gorkha earthquake, where strong agreement between inventories is the result of mapping from very similar satellite imagery.

Even when using a global parameter set, ALDI outperforms manual mapping in 8 of 14 cases (57%) with 10 of 14 cases (71%)
either performing better than manual mapping or within the uncertainty in manual mapping performance estimates. These results suggest that ALDI can be applied with considerable confidence to map the areas affected by co-seismic landslides in future earthquakes without the need for additional calibration. Holdback tests do not change either of these statistics and affect our chosen performance metrics by only a few percent, suggesting that the set of earthquakes that we have used is large enough to develop a robust global parameter set.

The area-frequency distributions for clusters of pixels that are classified as landslides both from manual and automated landslide classification are broadly similar, particularly in their heavy right tail. However, the classifier-derived inventories are fundamentally limited by the resolution of the imagery and their inability to disaggregate amalgamated landslides, so that an object-based approach is required to recover realistic area-frequency information.

ALDI is fast to run, uses free imagery with near-global coverage and generates landslide information that is of comparable
quality to that of costly and time-consuming manual mapping, depending on its intended use. Thus, even in its current form it has the potential to significantly improve the coverage and quantity of landslide inventories. However, its simplicity (performing only pixel-wise analysis) and parsimony of inputs (using only optical imagery) suggests that considerable further improvement should be possible.

*Code availability*

The Google Earth Engine code to run ALDI is available on Github (https://github.com/DavidMilledge/ALDI), and as a GEE App: https://dgmilledge.users.earthengine.app/view/aldi-landslide-detection

*Data availability.*

All data used in this research are openly available. The satellite imagery is provided by USGS and archived by Google within
Google Earth Engine. The Watt landslide inventory will be deposited in the USGS open repository of Earthquake-Triggered Ground-Failure Inventories on publication. All other landslide inventories used in this research are already in this repository.

*Author contributions.*

J.W. and D.G.M. collected one of the landslide inventories and made it ready for use. D.G.M. designed and implemented the
ALDI classifier with input from D.G.B. and analysed data with input from A.L.D.; D.G.M., D.G.B and A.L.D. wrote the paper. A.L.D. organized funds.

*Competing interests.* None

*Acknowledgements.*

Some of this work was undertaken while D.G.M. was supported by the Natural Environment Research Council [grants NE/J01995X/1 and NE/N012216/1]. D.G.B. was supported by a grant from the National Science Foundation (NSF EAR-1945431) and by a Gordon and Betty Moore Foundation Data-Driven Discovery Investigator Award (GMBF-4555). We are

extremely grateful to Google and the Google Earth Engine team for sharing their software, to the USGS for access to the
Landsat data, and to all the research teams involved in the USGS ScienceBase Landslide Inventory project for sharing their
landslide inventories. Comments from Odin Marc and two anonymous reviewers were very useful in helping us to refine our
approach and arguments.

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

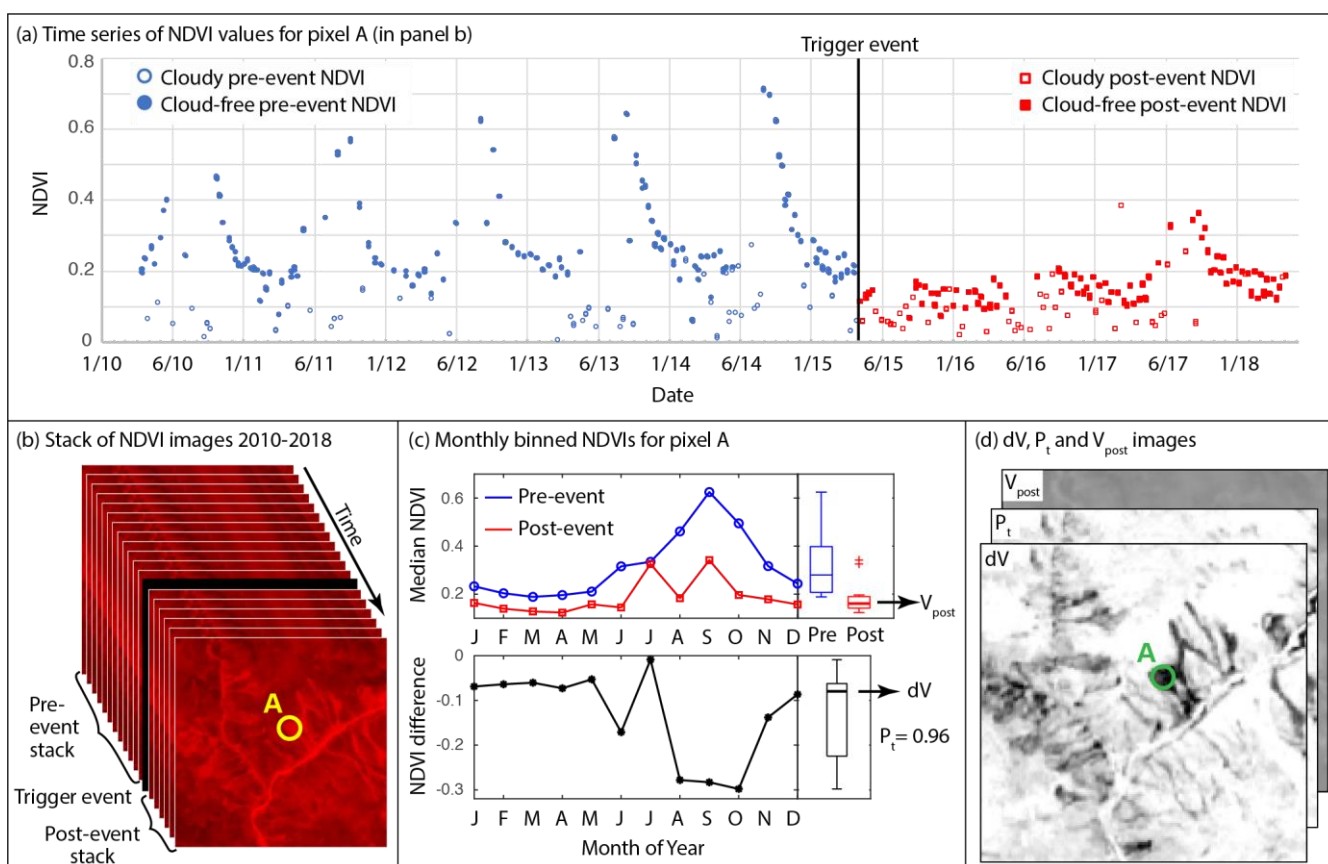

**Figure 1: ALDI pre-processing steps. (a) Time series of NDVI values for a single landslide-affected pixel (circled in panels b and d) before and after the trigger event, with cloud-free values shown as solid symbols. This time series is derived from a stack of NDVI images (b) and is used to calculate monthly median NDVI before and after the earthquake and their difference (c), which can be**
**used to calculate dV, $P_t$ and $V_{post}$ for every pixel in the study area (d).**

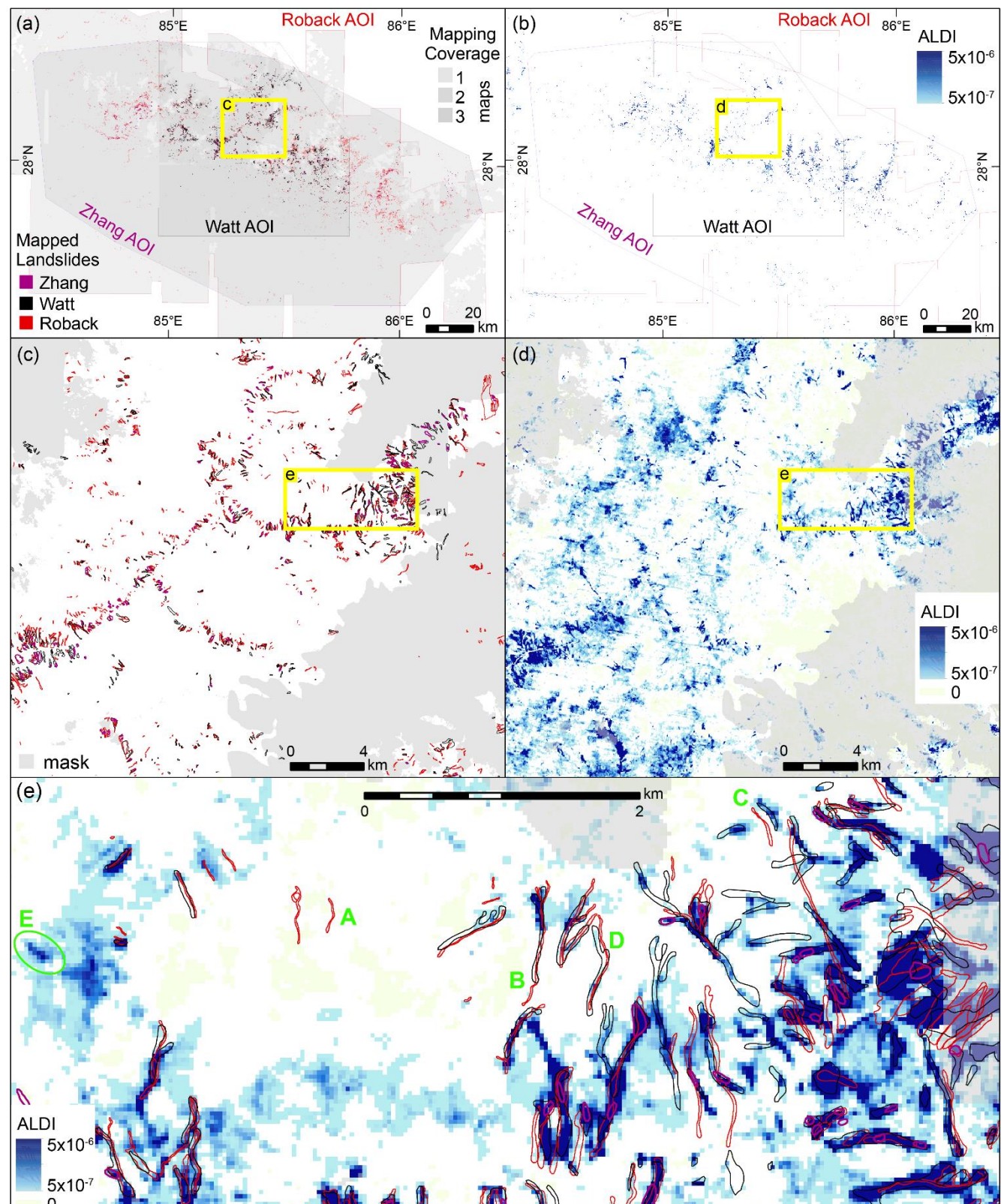

**Figure 2: Mapped landslides and the ALDI classifier for the Gorkha study site. a) Mapped landslides at the scale of the full study area with AOIs (the mapped area) shown in grey. Zhang, Roback and Watt refers to the inventories of Zhang et al. (2016), Roback et al. (2018), and Watt (2016). b) ALDI values for the full study area, using locally-optimised parameters. c) Mapped landslides from the three inventories for a subset of the study area, with areas that were unmapped in one or more inventory shaded grey. d) ALDI values using locally-optimised parameters for the same subset of the study area shown in (c); e) detailed view of mapped landslides from the three inventories and ALDI values. Yellow boxes in each panel show the locations of nested panels (e.g. (c) in (a) and (d) in (b)). Green labels indicate examples of: A) missed landslides, B) agreement between inventories, C) offset landslide outlines, D) distorted landslide outlines, and E) landslides identified by ALDI but missed by manual mapping.**

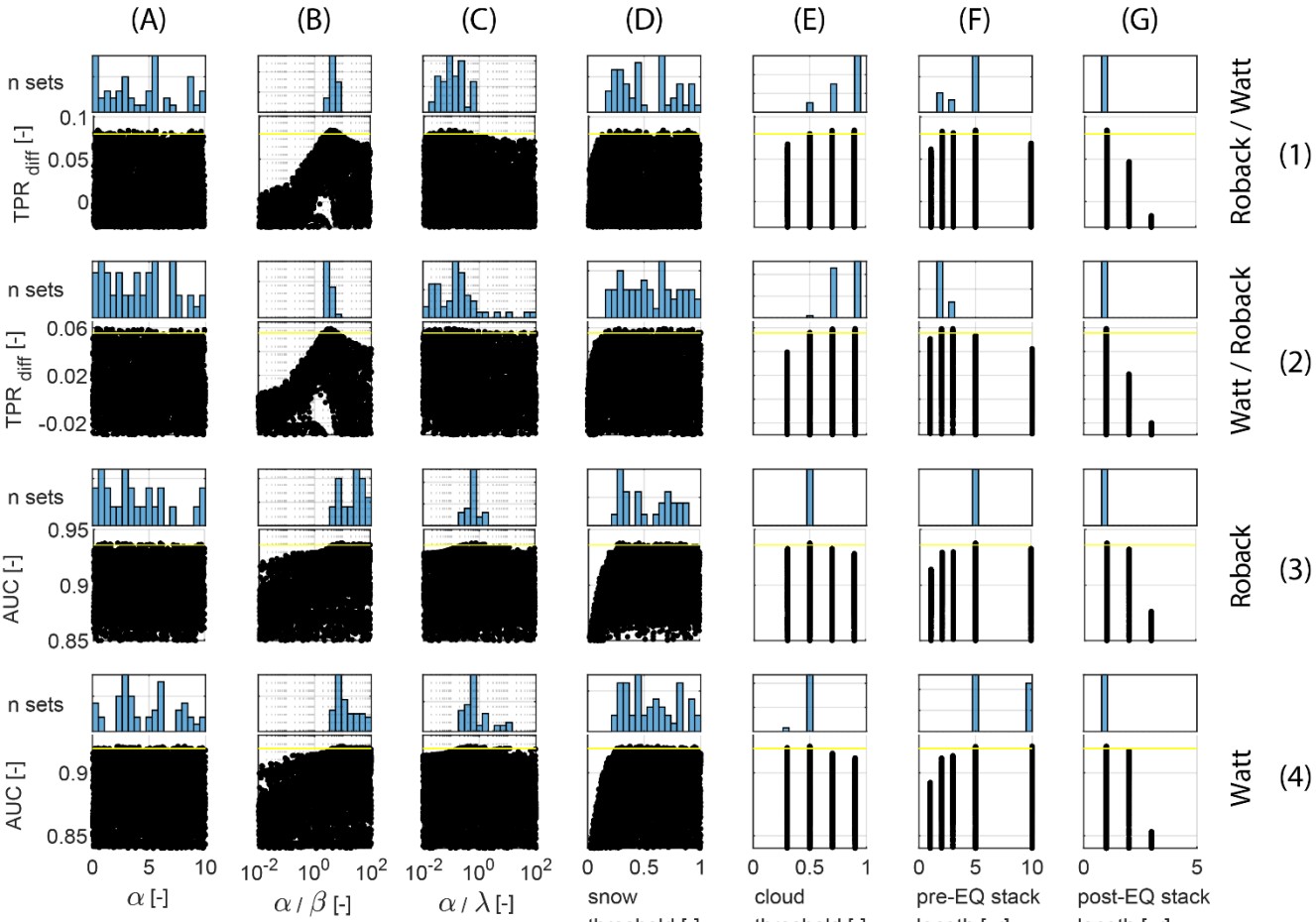

**Figure 3: Dotty plots and posterior parameter distributions for the Gorkha case study for the seven tuneable parameters associated with ALDI (columns) evaluated using two of the test datasets (Watt and Roback) and two performance metrics (rows): TPR$_{diff}$, the difference in TPR between ALDI and the competitor inventory at the FPR defined by the competitor inventory; and AUC, the area under the ROC curve, a more general indicator of classifier performance over the full range of FPRs. 'Roback/Watt' refers to using Roback as the check dataset and Watt as the competitor in row 1; 'Watt/Roback' refers to the converse in row 2. Roback is used as the check dataset in row 3, and Watt as the check dataset in row 4. Points plotting above the yellow line are results for the best 100 parameter values. In each case the parameter distributions are for the best 100 parameter sets evaluated using the same metric and datasets as the dotty plot below it. Dotty plots for the other Gorkha inventories and for all other sites are given in the Supplementary Information.**



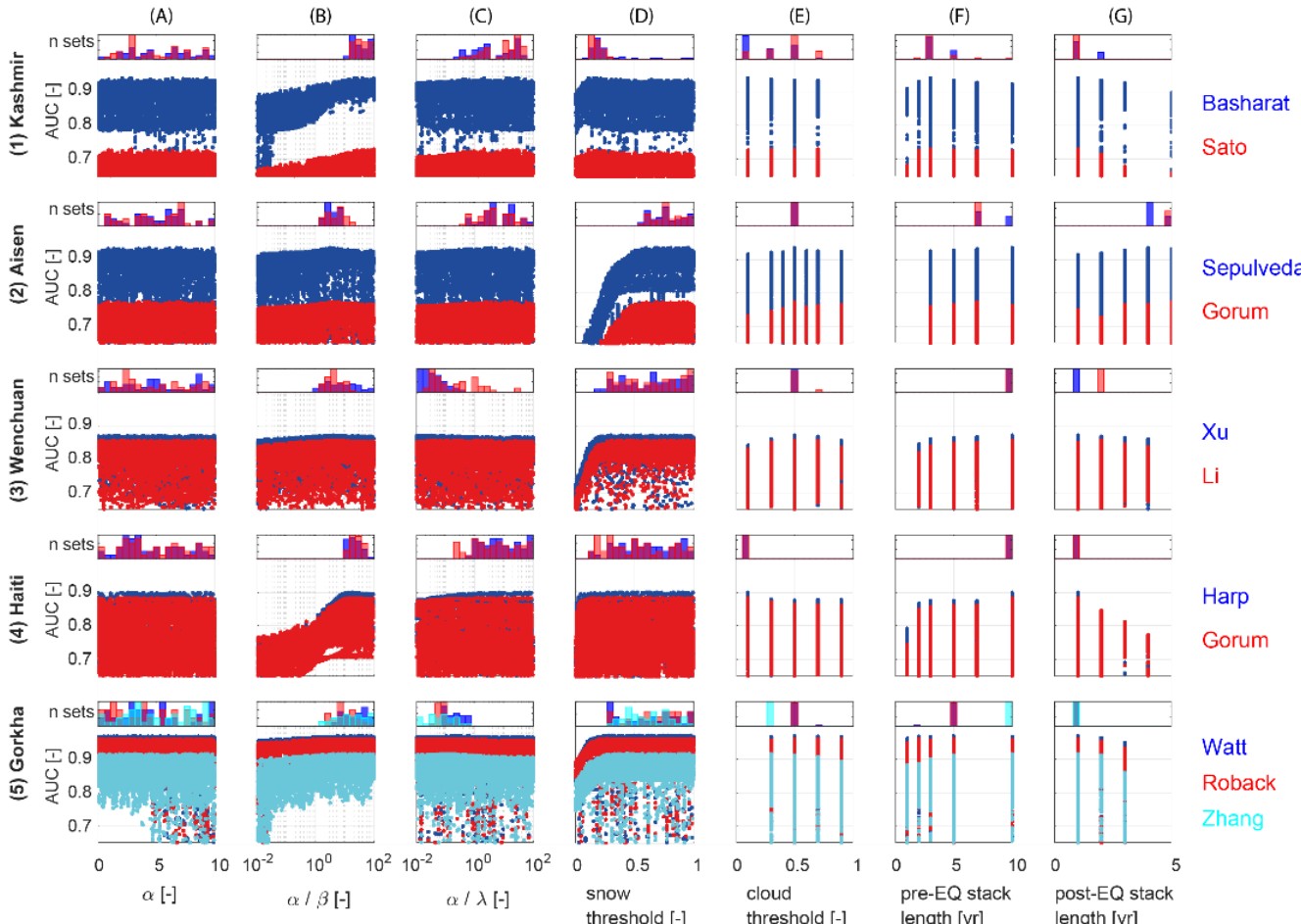

**Figure 4: Dotty plots and posterior parameter distributions for the seven tuneable parameters associated with ALDI (columns A-G) for the five study earthquakes (rows 1-5). Dotty plots show classifier performance evaluated using AUC, the area under the ROC curve. Blue or red colours indicate the inventory used as the check dataset, as shown to the right. Parameter distributions are for the best 100 parameter sets evaluated using the same metric.**


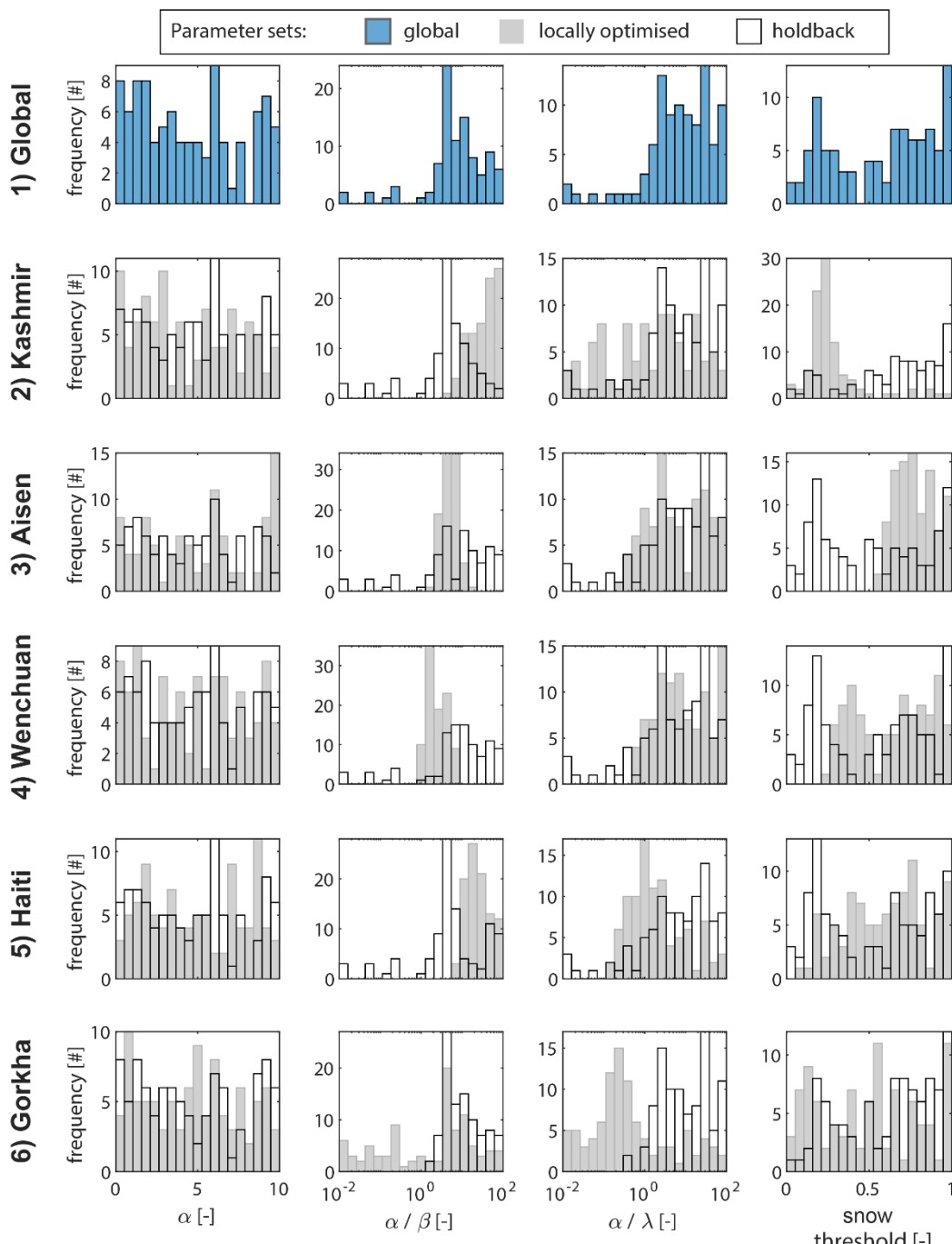

**Figure 5: Posterior parameter distributions for the four parameters external to Google Earth Engine after global optimisation (top row) and local optimisation for each earthquake. Rows 2-6 show posterior frequency distributions for each ALDI parameter following local optimisation (grey bars) and the holdback parameter set derived from the global set excluding locally optimised parameters (hollow bars).**


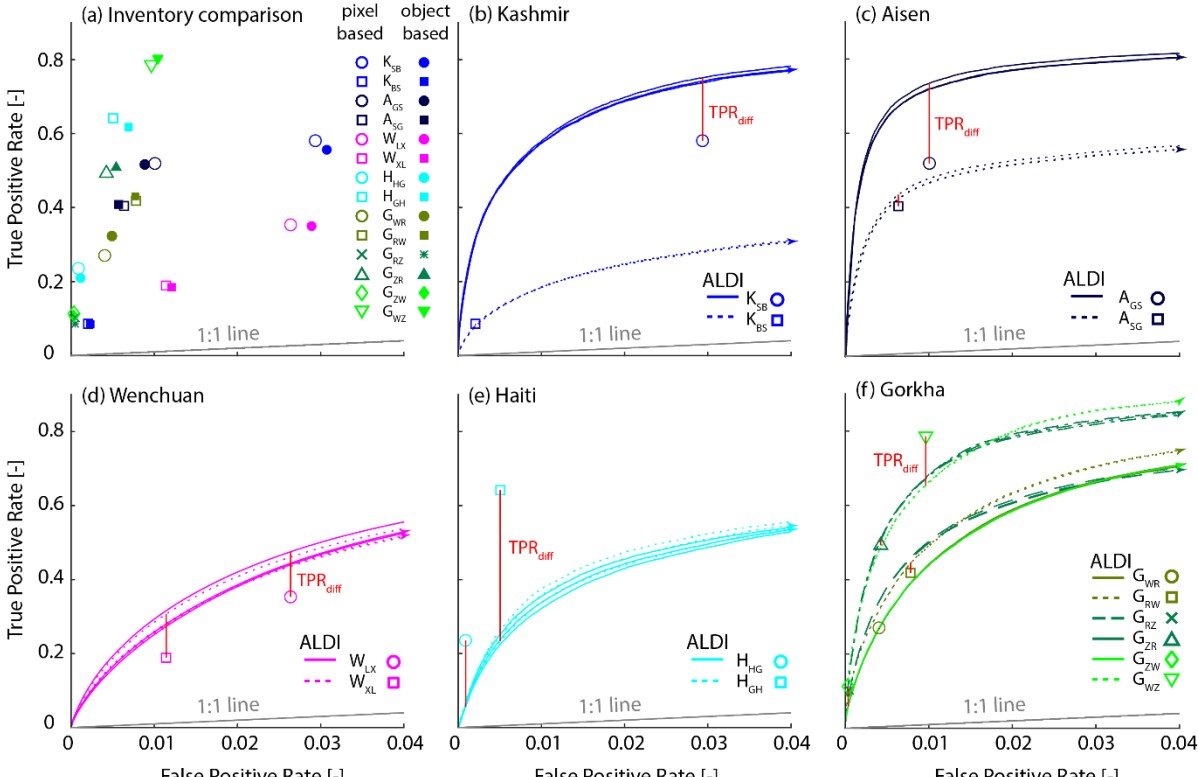

**Figure 6: a) TPR, FPR pairs for the 14 inventory cross comparisons. Open symbols are calculated from a pixel-based analysis at 30 m resolution, solid symbols are calculated from an object-based analysis using mapped polygons. The grey line shows the naïve (random) 1:1 relationship. Note difference in x- and y-axis scales for this and all other panels; b)-f) ROC curves for ALDI for each case study. There are three ROC curves for ALDI evaluated against each check inventory (e.g., $K_{SB}$) all with the same line style (solid or dashed). In every case the upper curve is from ALDI with locally optimised parameters, the middle curve (indicated with an arrowed end) is from ALDI with global parameters and the lower curve is from ALDI with holdback parameters. The global and holdback curves are indistinguishable in almost all cases. Red lines indicate the value of $TPR_{diff}$, the difference in TPR between ALDI and the competitor inventory when both are evaluated using the same check inventory. Legend acronyms indicate the study site (e.g., K) with the check and then competitor inventory labels as subscripts; see Table 3.**

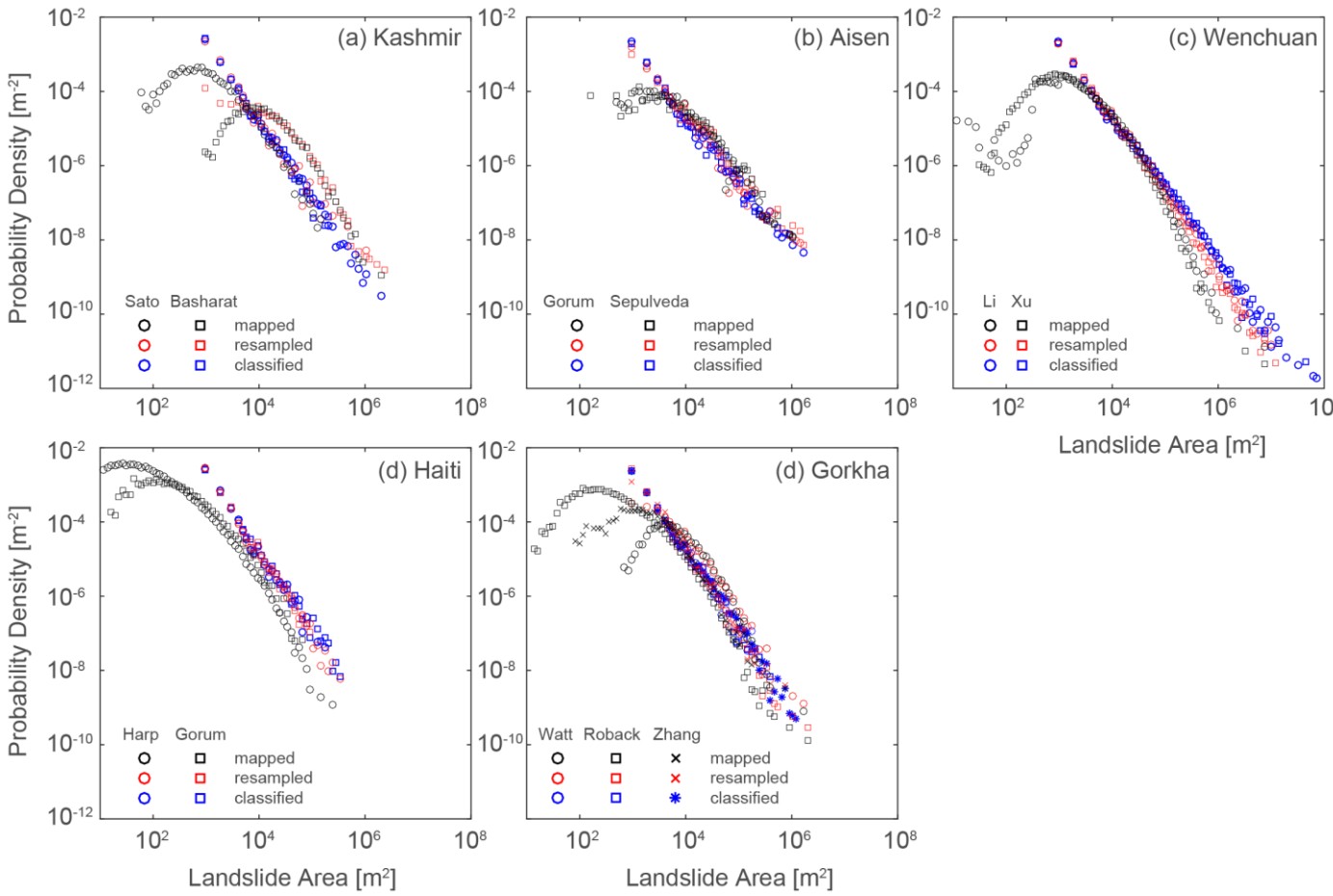

**Figure 7: Empirical area-frequency distributions for manually-mapped and classified landslides for the five case studies. Manually-mapped pdfs are calculated from areas of mapped polygons, resampled pdfs are calculated from patch areas generated from the mapped polygons resampled to a 30 m grid, and classified pdfs are calculated from clustered pixel areas generated by thresholding the ALDI classification values.**

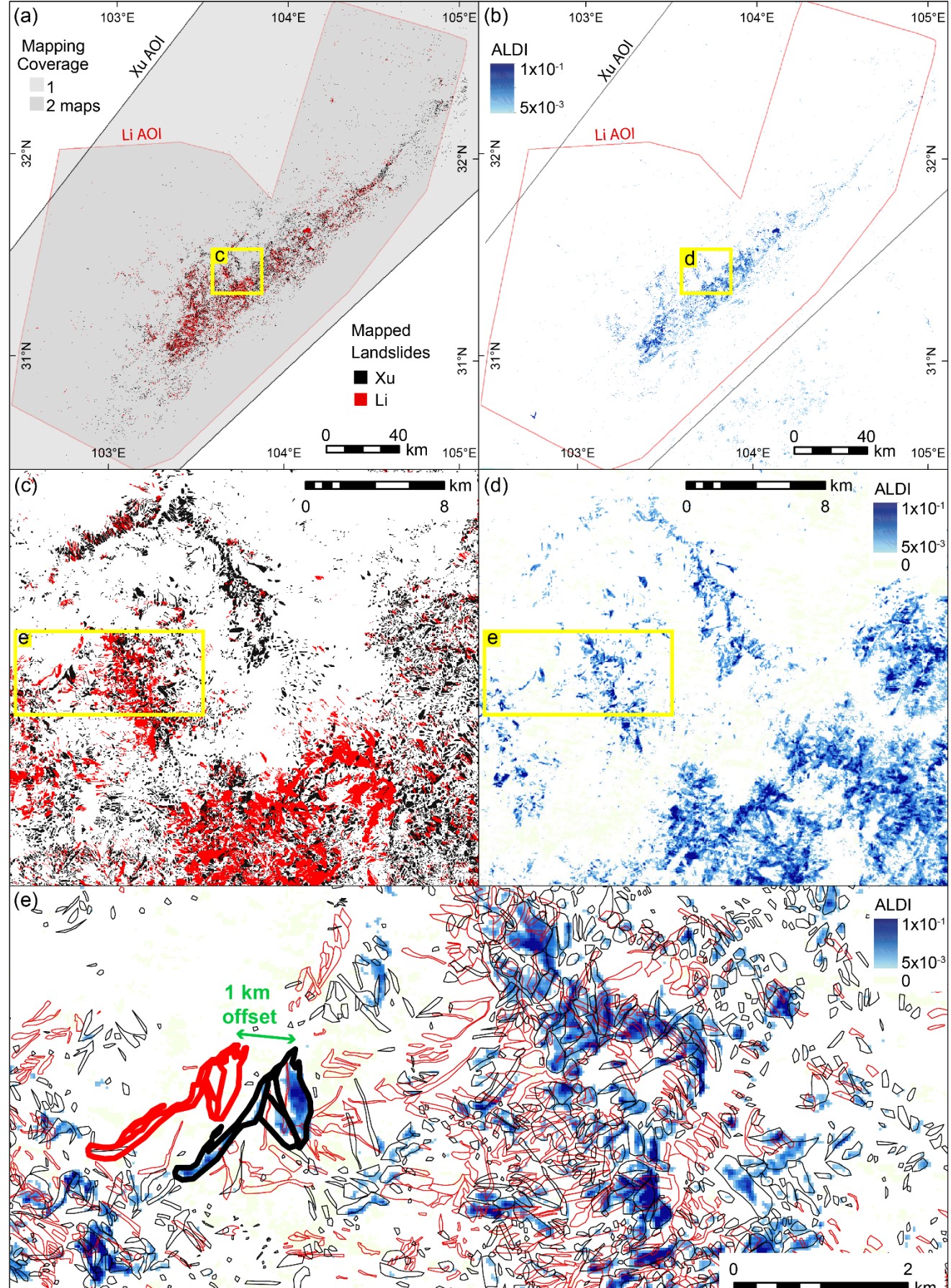

Figure 8: Manually mapped landslides and ALDI classifier results for the Wenchuan study site. a) Mapped landslides at the scale of the full study area with AOIs shown in grey, Xu and Li refer to the inventories of Xu et al. (2014) and Li et al. (2014) respectively; b) ALDI values for the full study area; c) mapped landslides and d) ALDI values for a subset of the study area; e) detailed view of mapped landslides overlain on ALDI values. Yellow boxes in each panel show the locations of nested panels (e.g. (c) in (a) and (d) in (b)). Thicker outlines in (e) indicate landslides of very similar geometry that are offset by ~1 km in the different inventories; the ALDI pattern suggests that the map by Xu et al. (2014) is more likely to be correctly georeferenced in this case.

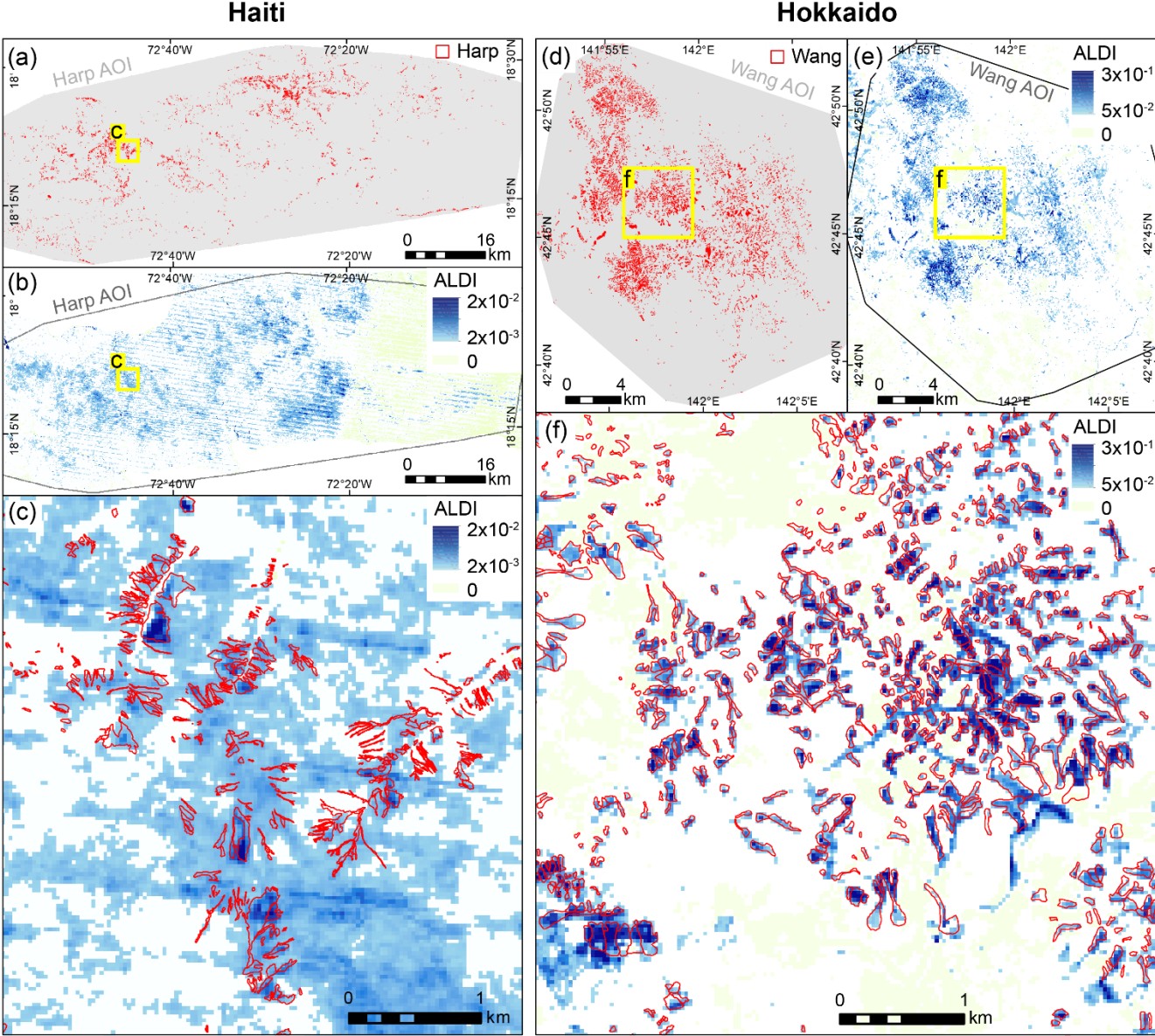

**Figure 9: Mapped landslides and the ALDI classifier for the Haiti (left) and Hokkaido (right) study sites. a) Mapped landslides from Harp et al. (2016) in Haiti at the scale of the full study area with the associated AOI shown in grey; b) ALDI values for the full study area, the yellow box shows the location of panel c; c) ALDI values overlain by mapped landslides from Harp et al. (2016) for a subset of the study area; d) Mapped landslides from Wang et al. (2019) in Hokkaido at the scale of the full study area with the associated AOI shown in grey; e) ALDI values for the full study area. The yellow box shows the location of panel f; f) ALDI values overlain by mapped landslides from Wang et al. (2019) for a subset of the study area. ALDI uses Landsat 5 and Landsat 7 for Haiti and Sentinel 2 for Hokkaido, both gridded at 30 m resolution.**

*Table 1: Landsat and Sentinel image characteristics (Barsi et al., 2014; ESA. 2017b).*

| | Landsat 5 and 7 | Landsat 8 | Sentinel 2 |
|---|---|---|---|
| Green (μm) | Band 2: 0.52-0.60 | Band 3: 0.53-0.59 | Band 3: 0.52-0.60 |
| Red (μm) | Band 3: 0.63-0.69 | Band 4: 0.64-0.67 | Band 4: 0.65-0.69 |
| Near infra-red (μm) | Band 4: 0.77-0.90 | Band 5: 0.85-0.88 | Band 8: 0.76-0.91 |
| Shortwave infra-red (μm) | Band 5: 1.55-1.75 | Band 6: 1.57-1.65 | Band 11: 1.51-1.70 |
| Spatial resolution (m) | 30 | 30 | 10 |
| Revisit time (days) | 16 | 16 | 5 |
| Operational life | 1984-2013 (L5) | 2013-present | June 2015-present (S2a) |
| | 1999-present (L7) | | March 2017-present (S2b) |

*Table 2: Parameters for Landsat simple cloudscore, Equations 4a-f*

| Threshold | Minimum | Maximum |
|---|---|---|
| Blue (Eqn 4a) | $R_{bmin} = 0.1$ | $R_{bmax} = 0.3$ |
| Visible (Eqn 4b) | $R_{vmin} = 0.2$ | $R_{vmax} = 0.8$ |
| Infra-red (Eqn 4c) | $R_{irmin} = 0.3$ | $R_{irmax} = 0.8$ |
| Temperature (Eqn 4d) | $R_{tmin} = 290$ | $R_{tmax} = 300$ |
| NDSI (Eqn 4e) | $NDSI_{min} = 0.6$ | $NDSI_{max} = 0.8$ |

1120

*Table 3: Performance metrics for ALDI applied with the different parameter sets to identify landslide-affected areas from each of the 14 inventory pairs. Abbreviated names for the inventory pairs indicate the case study with subscripts denoting first check and then competitor inventories (e.g., $K_{SB}$ denotes the Kashmir earthquake with Sato as the check inventory and Basharat as the competitor inventory). True positive rate (TPR) and false positive rate (FPR) are reported for both object-based analysis (in brackets), and pixel-based analysis at 30 m resolution. Overlap indicates the percentage overlap between pairs of landslide inventories. Shading in right-hand columns indicates performance of ALDI relative to each competitor and for each metric and calibration, with linear colour scale from blue where ALDI out-performs the manual competitor to red where the manual competitor out-performs ALDI. Vertical blocks reflect different performance metrics: $TPR_{diff}$ and AUC (see text). Columns within each block reflect different ALDI calibration strategies: local calibration optimised to both site and check inventory; global calibration using a compilation of the best parameter sets from all sites; and holdback calibration where parameter sets from the test site are excluded. Note that positive values of $TPR_{diff}$ reflect cases where ALDI out-performs manual mapping while negative values reflect cases where manual mapping is better.*

| Check Inventory | Competitor Inventory | TPR [-] Pixel-based (Object-based) | FPR [-] Pixel-based (Object-Based) | Overlap [%] | TPR$_{diff}$ [%] Local | Global | Holdback | AUC [-] Local | Global | Holdback |
|---|---|---|---|---|---|---|---|---|---|---|
| **Kashmir (K)** | | | | | | | | | | |
| ($K_{SB}$) Sato et al. (2007) | Basharat et al. (2016) | 0.58 (0.56) | 0.029 (0.030) | 8.2 | 30 | 26 | 27 | 0.94 | 0.93 | 0.93 |
| ($K_{BS}$) Basharat et al. (2016) | Sato et al. (2007) | 0.09 (0.09) | 0.002 (0.002) | | 0.2 | -7 | -5 | 0.72 | 0.69 | 0.69 |
| **Aisen (A)** | | | | | | | | | | |
| ($A_{GS}$) Gorum et al. (2014) | Sepulveda et al. (2010) | 0.52 (0.52) | 0.010 (0.009) | 29.7 | 56 | 39 | 39 | 0.93 | 0.93 | 0.93 |
| ($A_{SG}$) Sepulveda et al. (2010) | Gorum et al. (2014) | 0.4 (0.41) | 0.006 (0.006) | | 6 | 5 | 5 | 0.77 | 0.78 | 0.78 |
| **Wenchuan (W)** | | | | | | | | | | |
| ($W_{LX}$) Li et al. (2014) | Xu et al. (2014) | 0.35 (0.35) | 0.026 (0.029) | 14.0 | 36 | 26 | 27 | 0.87 | 0.85 | 0.85 |
| ($W_{XL}$) Xu et al. (2014) | Li et al. (2014) | 0.19 (0.19) | 0.011 (0.012) | | 62 | 50 | 51 | 0.86 | 0.84 | 0.84 |
| **Haiti (H)** | | | | | | | | | | |
| ($H_{HG}$) Harp et al. (2016) | Gorum et al. (2013) | 0.24 (0.21) | 0.001 (0.001) | 18.8 | -51 | -74 | -73 | 0.88 | 0.84 | 0.84 |
| ($H_{GH}$) Gorum et al. (2013) | Harp et al. (2016) | 0.64 (0.62) | 0.005 (0.007) | | -52 | -62 | -60 | 0.9 | 0.83 | 0.83 |
| **Gorkha (G)** | | | | | | | | | | |
| ($G_{WR}$) Watt (2016) | Roback et al. (2018) | 0.27 (0.33) | 0.004 (0.005) | 22.8 | 22 | 1 | 1 | 0.92 | 0.92 | 0.92 |
| ($G_{RW}$) Roback et al. (2018) | Watt (2016) | 0.42 (0.43) | 0.008 (0.008) | | 20 | 7 | 6 | 0.94 | 0.93 | 0.93 |
| ($G_{RZ}$) Roback et al. (2018) | Zhang et al. (2016) | 0.1 (0.09) | 0.001 (0.001) | 8.3 | 30 | -4 | -3 | 0.92 | 0.90 | 0.90 |
| ($G_{ZR}$) Zhang et al. (2016) | Roback et al. (2018) | 0.49 (0.51) | 0.004 (0.005) | | 19 | 4 | 5 | 0.96 | 0.95 | 0.95 |
| ($G_{ZW}$) Zhang et al. (2016) | Watt (2016) | 0.11 (0.11) | .0003 (.0003) | 11.1 | -28 | -47 | -47 | 0.92 | 0.92 | 0.92 |
| ($G_{WZ}$) Watt (2016) | Zhang et al. (2016) | 0.79 (0.80) | 0.010 (0.010) | | -9 | -17 | -17 | 0.97 | 0.97 | 0.97 |
| | Median | 0.38 | 0.006 | 14.0 | 20 | 3 | 3 | 0.92 | 0.91 | 0.91 |
| | Mean | 0.37 | 0.008 | 16.1 | 10 | -4 | -3 | 0.89 | 0.88 | 0.88 |