# Peer review of "Automated determination of landslide locations after large trigger events: advantages and disadvantages compared to manual mapping"

_Natural Hazards and Earth System Sciences, 2021_

## Author Comment (AC1)

**Reviewer 1**

Thank you for your thoughtful and detailed review. Your comments are much appreciated and we are convinced that they have helped us to considerably improve the manuscript. We hope that the responses below, together with the modifications in the manuscript address all your concerns. We have responded to each of your comments (in bold) below.

**MC1.1: The writing is per se quite clear but I'm to say that I had to read the paper several times to pull all the things together. The description of some steps arrives too fragmented, making the reader somehow supposing what will be next or giving the reader the possibility of taking wrong directions/expectations (e.g. I had to wait almost till the end to understand what the 'trade-off' of the time series was and how the length was selected, and the connection with natural processes).**

The reviewer's concern here is that the material is introduced in a fragmented way risking misinterpretation or allowing them to develop incorrect expectations. To address this general point we have thoroughly edited the paper to make our explanations more coherent and to clarify the structure. We deal with the reviewer's specific example (explanation of time series length) in detail where it is raised again in minor comments below.

**MC1.2. 3.1 is maybe too generic in the introduction of the algorithm, in how to prepare the time series, what the distributions are, and that a fitting method is used to find the parameters (they sound to me all parts of the method). I suggest trying & anticipate some concepts.**

This section started from very basic concepts and we appreciate the reviewer's advice to expect that the reader already knows some of this background (or that they can easily find it out). We have now modified the section, removing some of this unnecessary background and instead focusing on the algorithm from the start of the section.

**MC1.3. I also think that in the first 4 paragraphs some of the preparatory steps, comments, results, and interpretations (see detailed comments) are 'one way, not necessarily wrong but too biased to demonstrate the outperforming of the automatic method. In the discussion, this is a bit relaxed.**

We believe the reviewer was concerned about a potential bias in the paper, with the first four paragraphs as an example, and other examples picked up in detailed comments below. Specifically the reviewer's concern was that prior to the discussion we implied that outperformance in relation to one metric equated to outperformance in general. The bias was not intentional and we are thankful for the opportunity to clarify our writing. We have done so by: 1) specifying the key characteristics of landslide inventories and associated metrics of inventory quality in the introduction, and specifically differentiating between prediction of landslide location and geometry (L58-61 and L132-135); 2) introducing both location and size metrics within the 'performance metrics' section of the methods; 3) examining both metrics individually in the results, and then in Section 5.4 of the discussion clearly stating that ALADIN is comparable to manual mapping for location metrics but is worse with respect to size; 4) summarising findings in relation to the full set of metrics that we have examined in the abstract.

**MC1.4. In fact, the basic assumption that manual mapping is accepted as the most accurate method to map landslides is taken in a too broad sense and it is not critically reviewed neither contextualised.**

Rather than argue whether manual mapping is most accurate (or perceived to be so) we have revised the section to frame manual mapping as the most common method. There is a distinct lack of studies in the literature comparing manual and automated mapping and we see this as one of the contributions of this paper.

Previous text:

*"There have been some attempts at automated landslide detection from Landsat (e.g., Barlow et al., 2003; Martin and Franklin, 2005). The results of automated detection algorithms have not typically been framed as a viable alternative to manual mapping, however, but instead have been compared to a manual map of landslides that is*

*assumed to be more accurate and considered to represent the 'ground truth' (van Westen et al., 2006; Guzzetti et al., 2012; Pawłuszek et al., 2017). Automated or hybrid approaches still need visual interpretation for calibration, sometimes over large areas (e.g., Đuric et al., 2017) and there remains a perception in the landslide community that such techniques are neither necessarily more accurate (Guzzetti et al., 2012; Pawłuszek et al., 2017) nor less time consuming (Santangelo et al., 2015; Fan et al., 2019) than manual interpretation. Given the considerable investment of time and money involved in compiling an inventory, researchers continue to take a conservative approach and map by hand. It would therefore be useful to evaluate both automated classification and manual mapping against a common measure of performance."*

Rephrased to:

*"There have been some attempts at automated landslide detection from Landsat (e.g., Barlow et al., 2003; Martin and Franklin, 2005). However, manual mapping remains the most common approach to map landslides despite the time costs associated with it. Automated or hybrid approaches still need visual interpretation for calibration, sometimes over large areas (e.g., Đuric et al., 2017) and are typically compared to a manual map of landslides that is considered to represent the 'ground truth' (van Westen et al., 2006; Guzzetti et al., 2012; Pawłuszek et al., 2017; Bernard et al., 2021). There remains a perception in the landslide community that automated methods are neither necessarily more accurate (Guzzetti et al., 2012; Pawłuszek et al., 2017) nor less time consuming (Santangelo et al., 2015; Fan et al., 2019) than manual interpretation. Given the considerable investment of time and money involved in compiling an inventory, many researchers continue to generate inventories through manual mapping. It is therefore timely and useful to evaluate both automated classification and manual mapping against a common measure of performance."*

**MC1.5. The preference is in most of the paper given to the automatic mapping considering only some performance indices, but it does not take into consideration elements like the purposes of producing landslide inventories (in particular just after an event), the time needed to have long and adequate temporal series of satellite images to stabilize the signal in ALDI (at least one year if the sampling is consistent). The inability of the method to trap correctly small landslides is shown as a very secondary aspect, and the fact that, despite their presumed low qualities, manual inventories were used to tune the model (also the general one) is not remarked (without them ALDI could not be tuned). This is, as correctly stated by the authors, without a real benchmark**

This is a very good point, and we are thankful for the suggestion to consider the purposes of producing landslide inventories. We have added a few sentences in the introduction to define and explore these purposes and two paragraphs in the discussion (Section 5.4) to examine which purposes ALADIN would be more or less suited to.

The reviewer identifies a two of important limitations to the scope within which ALADIN can be used. First, the minimum mappable landslide size is still quite large (900 m$^2$); this was already discussed in the limitations section (Section 5.3) but is now introduced more clearly in the description of size-frequency results and re-visited in (Section 5.4). Second, ALADIN requires more than two years to have passed since the event, precluding its use in disaster response. We now emphasise this in our discussion of potential applications in Section 5.4.

Finally, the reviewer points out that the method has been trained using manually mapped inventories, albeit with a treatment of their uncertainty. However, we note that our global model can now be applied without the need for further manual mapping (i.e. no training data required) because the sites tested to date provide parameter sets that can be used 'blind' at new locations. We tested this 'blind' application using a holdback test and found that our conclusions hold even when global or holdback parameter sets are used.

**MC1.6. Nevertheless, I see some potentialities in the method (when better contextualised, and without unbalanced comparisons) to say that, given some inventories, it is possible to run it to update, or extend, or give homogeneity to the preexisting inventories (after many years of data acquisition), indicating the way of correctly using this type of product (for sure not in an emergency since it takes years to have the post-event time series).**

We thank the reviewer for their suggestions and have added text in the discussion to expand on these potential applications and to highlight areas where manual mapping remains more appropriate. We have sought to address the reviewer's concern that the method be 'better contextualised' by explaining more explicitly when ALADIN maps would or would not be useful (Section 5.3 on limitations, and 5.4 on performance of ALADIN relative to manual mapping). We have sought to address the reviewer's concern around unbalanced comparisons by providing a more balanced evaluation of both landslide location and size detection in the revised manuscript.

**In the detailed comments, I raise some issues related to some methodological steps that should be better explained or clarified.**

Thank you for these helpful comments.

**MC1.7. Last, some of the elements in fig 4 (distributions), 8, and 9 are very difficult to catch.**

We have edited these figures for clarity, and we have included descriptions of the key features in the text, so that the reader can understand our interpretations. We are not sure what specific elements the reviewer is troubled by, but we are happy to revise the figures further if more detail can be provided.

**MC1.8. I recommend for major review, and I strongly suggest the authors for a more adequate and multi-perspective contextualisation (maybe starting from the title, the outperformance is not absolute, but eventually relative to some choices).**

We agree that the outperformance is not absolute. In some respects manual mapping out-performs ALADIN, but in others ALADIN outperforms manual mapping. In addition, as discussed above, performance in reproducing landslide location is distinct from performance in reproducing landslide size or geometry. The title is a point of concern for both reviewers and we have sought to relax the title slightly to reflect their concerns. In particular, we believe that the reviewers' concerns are primarily that the term 'outperforms' needs contextualising. There is not sufficient space to do this in the title so we instead choose a more general summary of the paper's content.

*"Automated determination of landslide locations after large trigger events: advantages and disadvantages compared to manual mapping"*

The new title also reflects what we believe to be a more balanced examination of the performance of manual and automated mapping recognising the contexts in which one or other might have an advantage. We thank the reviewer for their recommendations, both major and minor, which have prompted a thorough revision of the paper.

**Introduction**

**115 – 125 address the problem that we don't currently have an objective comparison of manual mapping with automated classification: what is the meaning of objective here? Unbiased?**

We meant objective in the sense that there is no other 'ground truth' against which we can compare both methods we agree that the term was problematic and have re-phrased the paragraph to avoid using it.

**3.1 ALDI classifier: theory**

**170 – 175 reduces .. increases..: I understand the sense of the sentence but I don't think it is formally correct, I suggest something like: reflection in vegetated areas is lower/higher than in bare soil because…**

This text was part of a section identified by the reviewer as unnecessary and has been removed.

**180 – 185 negative: since there are many ways to measure changes in NDVI, I suggest making clear that here the difference is used.**

Agreed, we now say: *"We call the difference in NDVI before and after the trigger event dV, …"*

**180 – 185 regrow slowly (years..): not completely true for areas in south-est Asia. Maybe slowly enough with respect to the availability of the first images after the event... I think this is quite well explained in the next sentence, I suggest to try and connect the two.**

Agreed. Previous text: *"In addition, vegetated areas disturbed by landslides regrow slowly (over timescales of years to tens of years). Thus, for landslide affected pixels NDVI should not only reduce after the trigger event but also stay low for an extended period (at least one year)."*

Modified to: *"In addition, vegetation that is disturbed by landslides regrow slowly - over timescales of months to years (Restrepo et al., 2009). Thus, for landslide-affected pixels NDVI should not only reduce after the trigger event but also stay low for an extended period (at least one year, depending on climate and seasonality as well as the timing of the earthquake)."*

**194 systematically: is it the right term?**

You are right; periodically might be a better term here. The original sentence was: *"The time series are noisy because atmospheric conditions alter both incoming radiation (e.g., cloud shadow) and that received by the sensor and because ground surface (and especially vegetation) properties will vary over time both systematically (e.g., due to seasonal vegetation growth and harvesting) and randomly (e.g., due to leaf orientation)."*

*We have replaced "…both systematically (e.g., …" with "…both periodically (e.g., …"*

**195 – 205 Pt: a quite obscure part until at least the end of par 3.2, where it is possible to understand how to go from temporal series to distributions. I suggest finding a way to better introduce this concept. Furthermore, Pt can be high also for agricultural practices, or fires, also the state of the vegetation can be completely different.**

We agree that these other influences on NDVI are important. We now introduce the step from temporal series to distributions:

*"Since we expect NDVI to be noisy, we seek a third metric to identify whether there is a shift in NDVI in the presence of broadly consistent seasonal variations and random noise in NDVI. For this we take the difference in NDVI across monthly bins to account for the seasonal component, then quantify the shift in NDVI since the trigger event. For the shift to be indicative of real change it should be considerably larger than the noise present in the NDVI signal."*

We have also added two sentences to discuss the circumstances under which Pt can be high without landsliding:

*"High $P_t$ could also result from other events that reduce the coverage or vigour of vegetation, particularly if this involves complete removal (e.g., fire or logging). However, seasonal vegetation changes should be accounted for by examining monthly differences, while episodic events should only be noticeable when: 1) their timing is coincident with the earthquake and 2) their effect persists over more than one year."*

**210 -215 eq 2d: what is 'temperature'? What is the spatial res. Of this index?**

Our apologies, we had not properly defined our terms in Equations 4a-f. We now address this as follows:

*"where: $R_g$, and $R_b$, are the spectral reflectances from the red and blue bands; $R_{s1}$ and $R_{s2}$ are those from the first and second shortwave infra-red bands; and $R_t$ is that from the thermal infra-red band (the only band used here with a coarser 60 m resolution). The cloudscore parameters with min and max subscripts (e.g. $R_{bmin}$ and $R_{bmax}$ for the red band) are used to normalise pixel reflectances, their values are given in Table 2."*

**225 - 226 landslide probability: the probability of the presence of landslides in the images?**

Yes, your interpretation is correct. We have now clarified what we mean.

Previous text: *While this formulation is arbitrary, it has the advantage of allowing the index to take a minimum value of zero (indicating negligible probability of landsliding) if any of the individual terms is zero. Because we have no a priori knowledge of the relative importance of each parameter in determining landslide probability, we assume a power-functional form with empirical exponents $\alpha$, $\beta$ and $\lambda$:"*

We now say: "*(indicating negligible probability that the images reflect a landslide at that location)*" and "*… in determining the landslide signature*"

**ALDI does not seem to me so directly connected to a landslide probability unless we assume that Pt, the probability that two distributions are different can be called landslide probability (but then the H0 should be related to). Maybe in another way: is it enough to say that this is 'landslide probability'?**

We agree that this was confusing. The changes in response to the previous comment address the confusion.

**231 – 232 due to a large variance between pre-event and post-event NDVI distributions: see my previous comment at 195 – 205. I guess here the authors refer to the boxplot of fig. 1-c difference, if yes, it sounds to me very far from being normal distributed, so I would not use variance unless the distribution is modeled, and the distribution has a variance. If not, what are the distributions?**

Our apologies, variance was the wrong term to use here.

The previous text was: *"Landslide pixels should be characterised by negative dV, indicating vegetation removal; low $V_{post}$, indicating a lack of vegetation after the earthquake; and high $P_t$, due to a large variance between pre-event and post-event NDVI distributions."*

We have replaced it with the following text:

*"… and high $P_t$, due to a distinguishable shift in post-event NDVI distributions relative to the pre-event distributions."*

**3.2 ALDI classifier implementation and data pre-processing**

**250 – 255 trade off: actually it seems to me that the final/correct length of the time series is found according to a best fitting process and not related to a study of the physical processes that characterise the possible evolution of the time series. The only choice a priori is related to the length of the time series used in the fitting. This concept remains a bit misleading until the calibration, at least for me, because I was waiting for a characterisation of the processes to monitor in the different areas.**

You are correct, the stack lengths are calibrated to best fit the observations rather than defined from theory, though the calibrated values do sit within the range that we might expect from theory. Thank you for pointing out the risk of misleading the reader early on. We have tried to address this (with minimum disruption to the flow of earlier text) by pointing this out and pointing to the upcoming calibration section in an additional sentence at the end of the paragraph:

*"These parameters are found using the calibration process described in Section 3.4 rather than by considering the physical processes that characterise the possible evolution of the time series.*

**A curiosity about the sample size: any limit imposed by Google?**

We have not experienced any sample size limit to date but our focus has been on individual earthquakes, which has set the scale of our study areas.

**259 for each pixel: in the satellite image?**

This is now explained - *"for each pixel in the pre- and post-earthquake stacks"*

**260 - 265 short enough .. long enough …: (similar, and connected to some previous comments)**

We believe that we have dealt with this comment in our response and revisions to previous comments.

**Generally: does it mean that some remain empty? If yes, how many? How do you cope with this?**

Yes, some remain empty. Any monthly bin that does not contain data in both the pre- and post-earthquake period is neglected in the subsequent analysis. We now say:

*"Monthly bins result in four images per bin per year on average, and thus empty bins are very unlikely except for month-location pairs that are characterised by extreme cloudiness (such as Nepal in July; see Wilson et al., 2016). Monthly bins that are empty in either pre- or post-earthquake period are not used in the subsequent analysis, with calculations for that pixel performed using the remaining monthly bins."*

**Short enough to capture seasonal changes: please add references that this is true for the test areas. Furthermore, is the irregular initial sampling not influencing the result (see Nyquist)?**

We have clarified the text by emphasising that we are capturing the annual pattern with reference to Figure 1 to illustrate this point: *"Monthly bins are used since they are generally long enough to contain data in every bin (even after removal of cloudy pixels) but short enough to capture annual seasonality (e.g. Figure 1a)."*

We agree that our sampling can be irregular because of the removal of cloudy pixels. This could cause problems if were aiming to reconstruct the timeseries. Instead, to calculate $P_t$ we transform the timeseries by binning then averaging so that annual cycles dominate, enabling the comparison at similar times of year. We note that $dV$ and $V_{post}$ are robust to irregular sampling since they are averages over multiple years.

**What does more robust to outliers mean here? How were outliers identified (what test and on what type of distribution)?**

We meant that the median is less sensitive than the mean to skew and extreme values. Outliers was not the right term here because we do not define the distribution. The previous text: *"We calculate median NDVI for each monthly bin, choosing median rather than mean since it is more robust to outliers (Figure 1c)."* has been modified to: *"We calculate median NDVI for each monthly bin, choosing median rather than mean since it is less sensitive to skew and to extreme values (Figure 1c)."*

**265 Vpost: I'm confused, Vpost is not obtained from the distribution of the differences. Is it?**

You are right, thankyou for spotting this. Vpost is obtained as the mean of the post earthquake NDVI distribution. We have now addressed this by moving the definition of Vpost later in the paragraph.

*"We take the mean of the post-event monthly NDVI values to generate $V_{post}$."*

**265 – 266 t-test: assuming that the mean of the differences between the medians should be 0 (H0), it is also assumed that the distribution of the differences is approximately normal, something that I guess should be tested in the different situations, and in a robust way, including years with different rain seasons.**

We were not seeking to use the t-test to establish formal probabilities rather as an index of change that accounts for expected variability due to noise in NDVI over time. We now clarify this in the text:

*"Since we expect NDVI to be noisy, we seek a third metric to identify whether there is a shift in NDVI in the presence of broadly consistent seasonal variations and random noise in NDVI. For this we take the difference in NDVI across monthly bins to account for the seasonal component, then quantify the shift in NDVI since the trigger event. For the shift to be indicative of real change it should be considerably larger than the noise present in the NDVI signal. Thus, we express the NDVI shift relative to the noise for each pixel as.*

$$t = \sqrt{n}\frac{dV}{S_v} \tag{2}$$

*where n is the sample size (12 for monthly bins), dV is the mean of the monthly NDVI differences, and $S_v$ is the standard deviation of the monthly NDVI differences. We then normalise by mapping t onto the cumulative Student's t distribution to generate $P_t$, the likelihood that the pre- and post-event NDVIs are drawn from different distributions:*

$$P_t = I_{\frac{(n-1)}{n-1+t^2}}\left(\frac{n-1}{2},\frac{1}{2}\right) \tag{3}$$

*where $I_x(a,b)$ is the regularized incomplete beta function. While this is equivalent to a paired t-test, the results cannot be interpreted as formal probabilities, as the distribution of dV may not be Gaussian. Rather they represent an index of change relative to expected variability which is bounded by [0, 1]."*

Alternative approaches could retain a tighter connection to formal probabilities by using a non-parametric statistic. However, testing these alternatives in the current manuscript is not feasible because it would necessitate complete reprocessing of all datasets. This would be extremely time-consuming because in the analysis for this paper the ALADIN surfaces in Google Earth Engine are post-processed in ArcMap and Matlab, requiring manual export of each individual ALADIN image.

**I don't know how representative fig 1 is of all cases, but there is a 'constant' shift between the blue and the red curve before the event (mean different from 0). Is it a matter of different rainfall in the different years? Need to atmospheric correct the images?**

You are correct that there is a shift between red and blue curves in Figure 1a (red represents pre-event and blue post-event images) and that this is picked up in the monthly averages in Figure 1c (again red for pre- and blue for post-event monthly medians). The key point, which we now emphasise in the caption, is that this data is all for a single pixel that was affected by a landslide triggered in the Gorkha earthquake. Therefore we believe this shift to be the signal of NDVI change due to landslide disturbance of the vegetation in this pixel.

**3.3 Performance testing**

**Another general comment: I could not find the way the authors cope with the different resolutions/scales of the products in comparison and competition.**

We now explain in the methods section that: *"The manual map is then rasterized to the same resolution as the classification surface - in this case, 30 m - using a 'majority area' rule, whereby landslide pixels are those with the majority of their area overlapped by landslide polygons."* We chose 30 m resolution because that is the finest resolution at which ALADIN can be recovered without interpolation and because comparison at finer resolution would introduce extremely high computational demand particularly for the larger study areas (e.g. Wenchuan).

We have tested the influence of this resolution choice by examining the impact of rasterizing the landslide maps at a much finer (1 m) resolution then resampling the global average ALADIN surface at the same resolution. The analysis was extremely computationally demanding but the results suggested only minor reduction in performance insufficient to change the key conclusions on the relative performance of automated vs manual mapping.

We examined the impact on AUC and TPRdiff metrics with respect to resolution in each case. We chose Aisen as the study area because it is the smallest of those that we examined. However, even this study area was prohibitively large at 1 m resolution causing memory errors during ROC curve generation in Matlab. We thus evaluated the impact of resolution on 9 sub-grids of equal size. We repeated the process for the two manually mapped landslide inventories available for the site, generating 18 sets of observations.

We found that mean AUC was reduced by 0.0069 (or 1.4%) when finer (1 m) resolution landslide data were used. The variability between the 18 trials was fairly small with a standard deviation of 0.0253. We found that mean TPRdiff was reduced by 0.03 (or 14%) on average when finer resolution landslide data were used, with a standard deviation of 0.15. Importantly, the resolution change never altered the sign of TPRdiff in any trial, indicating that the magnitude of the change in each case was not sufficient to alter the conclusion about whether the manual or automated inventory better reproduced the check data.

**Furthermore, how good is the co-registration among the data?**

We have added the following statement on Landsat co-registration:

*"To ensure satisfactory image-to-image registration for time-series analysis we use only images which have been both georeferenced to ground control points and terrain corrected (i.e. Level 1TP) and thus have ≤12 m Radial Root Mean Square Error (RMSE) in >90% of cases (USGS, 2019)."*

Landslide inventory co-registration uncertainty is rarely reported and not reported for any of the inventories used here. However, our results suggest that co-registration errors are likely responsible for a considerable fraction of the disagreement between inventories. We have added details on georeferencing where possible and highlighted the lack of information for other landslide inventories:

*"The registration errors in the Watt (2016) were estimated from those associated with the underlying imagery from which the landslides were mapped. These Landsat 7 and 8 images were all geo-referenced to Level 1TP resulting in radial RMSE of <12 m (USGS, 2019), which is less than the pan-sharpened pixel resolution (15 m). We were unable to find registration error estimates for the other landslide inventories examined here."*

**275 relative measure of the confidence: do the authors mean statistical confidence here?**

No, we have replaced "confidence" with "certainty" to avoid statistical connotations.

**290 – 295: I suggest mentioning in which situation this choice of working with TPR and FPR is preferred. Here (https://doi.org/10.3390/rs12030346) for example, the authors make a different choice for rapid mapping.**

We now clarify that: *"The true and false positive rates are insensitive to imbalanced data and thus are well suited to evaluation of landslide classification, which typically has many more non-landslide than landslide pixels (García et al., 2010)."*

This insensitivity to imbalance is in contrast to the F1-score and Mathew's correlation coefficient (which are among the metrics used by Prakash et al., 2020). In addition, working with TPR and FPR, enables us to calculate TPRdiff, which we think is particularly useful in our case. We have added the following text to explain our choice:

*"This approach allows direct comparison between ALADIN and manual mapping for the same classification threshold. Other metrics could be derived from the confusion matrix (e.g. Tharwat, 2020; Prakash et al., 2020)*

*but these typically require assumptions about the relative weight assigned to true and false positives and negatives. Our approach avoids these assumptions because ALADIN is thresholded to ensure that FPRs are equal to those of the competitor inventory."*

**298 – 299 ALDI must first be thresholded to generate a binary classifier with the same FPR as the competitor inventory with respect to the check inventory: sorry not sure I understand this important step: Is ALDI calibrated using the competitor inventory to obtain the same FPR, correct?**

Yes, to compare ALADIN to the competitor inventory we choose to threshold the continuous ALADIN to generate a binary grid that has the same FPR as the competitor.

**If yes, it seems to me that the outperforming of the automatic mapping is (or can be) strictly related to this choice: once the FP is under control (usually the biggest problem in automatic landslide detection), then TP outperforms. How about tuning to make TPR(aldi) = TPR(comp)? Would the results be similar? I think the choice is quite 'smart' because it tends to mitigate one of the issues in automatic mapping, but it must be highlighted that the choice strongly depends on this a priori choice. Is this what the authors mean when they talk about weakness? If yes, I think this point should be better discussed and highlighted.**

As an alternative to enforcing the same FPR for both ALADIN and the competitor inventory we could have enforced equal TPRs for ALADIN and the competitor. We tested both and found little difference between the two. In particular, because ROC curves are monotonic, while the distance between ALADIN ROC curve and the competitor point (in ROC space) will differ depending on whether it is measured in x (i.e. FPR) or in y (i.e. TPR) the sign of the distance will not (i.e. the competitor is either above and to the left of the ALADIN curve - indicating that the competitior is more skilful; or it is below and to the right - indicating that ALADIN is more skillful).

**3.4 Parameter calibration and uncertainty estimation**

**320 – 325 working from the most to least sensitive parameter for each earthquake event and then checking for interaction between parameters: sorry, what does it mean? How was the sensitivity ranking established and how did the authors check for interactions?**

We have modified the text to clarify. It now reads: *"We calibrate $L_{post}$, $L_{pre}$, and $T_{cloud}$, one-at-a-time (in that order) for each earthquake event then test alternative near-optimum parameter combinations to minimise the effect of the calibration order. These combinations are obtained by varying $L_{post}$ by +/- one year for optimum values of $L_{pre}$ and $T_{cloud}$, doing the same for $L_{pre}$ at optimum values of $L_{post}$ and $T_{cloud}$."*

**Comment just for sake of discussion: since some of the parameters are exponents, a random sampling from a uniform distribution might not be able to trap the ranges in which the parameters landscape has a higher slope.**

You are correct, the sampling strategy for the exponents was a concern for us too. We tested this and modified our parameter choice and sampling to better cover the sample space. However, we apologise for an incomplete explanation of this exercise in the previous manuscript. We impose prior distributions of $\alpha$, $\alpha:\beta$ and $\alpha:\lambda$. We chose to impose these distributions as inputs rather than imposing $\beta$ and $\lambda$ then recovering the ratios because we found that ALADIN was sensitive to these ratio parameters but not the individual parameters. We sample $\alpha$ from a uniform distribution but sample uniform distributions of $\log10(\alpha:\beta)$ and $\log10(\alpha:\lambda)$. We chose to sample in log10 space for the ratio parameters in order to maintain symmetric sampling density with distance from a ratio of unity (e.g. $\alpha:\beta=0.1$ where $\alpha =10$ $\alpha:\beta$ should be sampled as densely as $\alpha:\beta =10$ where $\beta = 10* \alpha$). We have updated the text to make this clearer:

*"For each GEE run in the one-at-a-time process we run 500 simulations of Equation 6 with $T_{snow}$ and $\alpha$ randomly sampled from uniform probability distributions and the ratio parameters sampled from uniform distributions of $\log10(\alpha:\beta)$ and $\log10(\alpha:\lambda)$. We sample the ratio parameters in logarithmic space to maintain symmetric*

*sampling density with distance from a ratio of unity (e.g. α:β=0.1 where α=10β should be sampled as densely as α:β=10 where β=10α). We explore a wide range of parameter values to increase the likelihood of capturing the optimum performance."*

One final minor point worth noting is that we do not use a gradient climbing optimisation so the slope of the parameter landscape is less critical in our case.

**325 – 330: see my previous comment on the a-priori choice of the time series length. I suggest mentioning these criteria (not the numbers) before, in the methods paragraph. Still, I suggest listing the main 'other landscape changes that might occur in the different areas and references in which it is said that the temporal window and the frequency sampling are adequate. Maybe fires? Drought?**

We have addressed the previous comment mentioned here by pointing to this calibration section. We do not add further detail there because we think this text is necessary in the calibration section and we want to avoid repetition. We have added the main 'other landscape changes' and a reference for the timescale of scar revegetation.

*Adjusted text: "We examine $L_{post}$ of up to five years because vegetation typically begins to re-grow over this timescale (Restrepo et al., 2009), and $L_{pre}$ of up to ten years because we expect that other landscape changes (e.g. fire, drought and landslides caused by other triggers) will begin to disrupt the pre-event signal at longer timescales. In both cases we examine only integer year values to ensure consistent sampling within the monthly bins."*

**330 – 340 : sorry but the entire subparagraph is not fully clear to me, maybe just because I'm not an expert in it. "measured in terms of AUC": to obtain the ALDI threshold which gives the best AUC, or the threshold which gives FPR(aldi) = FPR(comp)?**

Original text: *"We then combine the best 20 parameter sets (measured in terms of AUC) from each earthquake into a global parameter set."*

It is the former, we find the 20 sets that give the best AUC for each of the five sites then combine them to give a group of 100 sets. We have changed the wording here to make this more clear. Note though that there is no thresholding involved here, AUC does not require that a threshold is defined. We use AUC because TPRdiff requires that we know something about competitors and we are trying to avoid giving any information about competitor inventories until they compete.

**combine: what does combine mean here? Or: how did the authors combine the sets?**

We find the best 20 sets for each site then we combine them to make a group of 100 sets. Ultimately we run the model for each of these 100 parameter sets to generate 100 ALADIN surfaces then take the mean of the 100 ALaDIn values for each cell.

We have clarified this by: 1) replacing "combine" with "retain" and 2) replacing "global parameter set" with "global set of 100 parameter sets"

**Parameter interaction: how do the parameters interact?**

The exponents interact strongly as we would expect from equation 6. The other parameters do not appear to interact as far as we can tell. We now say:

*"To account for parameter interaction (particularly between the three exponents: $\alpha$, $\beta$ and $\lambda$) within a set we retain parameter sets as 7-element vectors."*

**Equal weight: is it really necessary? In the end, the purpose is to get what gives the best final results.**

We believe it is necessary. The objective here is to generate global parameter set that might be used by others in new locations rather than to generate the parameter set that gives the best final results when evaluated with respect to our inventories. Site and inventory specific calibration has already identified the single best performing parameter set for each site-inventory pair and recorded the associated performance metrics.

**3.5 landslide size**

**General comment: I understand the purpose but I have the feeling that this is not a real comparison because resampling the manual inventories introduces 2 issues: 1) you are not more using the original product, 2) you are losing one of the main pros of the manual inventories related to the capacity of the operator to distinguish between small landslides,**

We apologise because we think this comment likely results from a misunderstanding of our method resulting from unclear explanation. We do compare the ALADIN size distributions with those from the manual inventories in their original vector form. You are absolutely right that resampling the manual inventories degrades the original product. We resampled the manual inventories to diagnose the source of the misfit between ALADIN and manual size distributions (but this is difficult to explain in the methods since it becomes clear only on seeing the results). We have now moved all discussion of the resampled manual inventory to the results to avoid this confusion.

**345 – 350 First...: the concept was somehow already introduced earlier.**

Fair point, though we aren't saying this is the first time we mention this, only that this is the first of a series of steps. We have modified the sentence to address the concern:

*"First, we generate a binary prediction of landslide presence or absence by thresholding the ALADIN classification surface to match the manually-mapped FPR, as described above."*

**345 - -350 current practice in landslide mapping: I repeat something already commented earlier. As far as I know, other authors prefer different strategies, according to the use that the map is devoted to. I suggest adding a reference to say that this is current practice.**

We mean that human mappers are typically conservative in their manual mapping only including features that they are convinced are landslides. We have not found an explicit statement to this effect in the literature though we believe that the consistently low FPRs found here for 11 inventories across five study areas support this claim. Given this, we have modified the text to explain our choice without making a wider claim:

*"… thresholding the ALADIN classification surface to match the manually-mapped FPR, as described above. The manual inventories examined here typically have very low FPRs (<2% of TPR on average and <7% at most, Table 3)."*

**4 Results**

**4.1 Spatial agreement: Gorkha case study**

**General comment of Fig. 2: It is not very easy to see the differences (actually I can't), maybe because the area is too large also in the windows c and d. or the quality of the figure I have in the pdf is too low. It would be nice to see some examples of mapped landslides in some VHR optical images (like 2e but without ALDI and more zoomed).**

We specifically show the results at the broad scale of the entire study area and two closer scales, to illustrate both the overall pattern of ALADIN results and the match/mismatch between ALADIN and the two inventories. We have made some minor alterations to Figures 2, 8 and 9 to simplify them (particularly at the full study area scale) and to improve their clarity. Overlaying the landslide inventories on high-resolution optical imagery

creates a very noisy figure that does not speak directly to the focus of this manuscript. We are happy to consider alternative ways of demonstrating the key results if the reviewer and editor would like to see further changes.

**Another point, I see some parallel lines in the ALDI results (not only in fig.2), a sort of high-frequency noise, what is it? Striping problems? Or aliasing?**

You are correct that you can see parallel lines in the results. Those lines are the result of a smaller sample size in these pixels because no data is available from Landsat 7 due to scan line correction failure. We have added the following text to highlight and explain the stripes.

*"Examining a subsection of the study area (Figure 2d) shows that ALADIN identifies the same broad zones of more intense landsliding as identified in the manual mapping. ALADIN also displays a series of stripes ~1 km apart and ~150 m wide trending west-northwest to east-southeast, however, and most clearly visible across the centre of the map. These are the result of data gaps in Landsat 7 images since 2003 due to Scan Line Corrector (SLC) failure on the Landsat 7 sensor. Although both pre- and post-event image stacks include Landsat 5 and 8 images in addition to Landsat 7, these data gaps clearly influence the ALADIN surface, with high values more likely for pixels where Landsat 7 data are not available."*

**370 – 375 a number of false positives in the south and west of the study area: 2 comments: I think they become false positive once the threshold is selected, 2) actually with AUC the threshold is selected to make FPR(aldi) = FPR(comp), so, most of them if not all should disappear. Sorry, I don't understand the sentence.**

The reviewer makes two good points here. First, that these patches only become false positives when a threshold is selected and none is selected here. The reviewer's second point is less clear to us. AUC, the area under the ROC curve does not involve selecting any threshold, which is one of its major advantages. The TPRdiff analysis, also uses the ROC curve but does require thresholding and as the reviewer points out the threshold is selected *"to make FPR(aldi) = FPR(comp)"*, in this case most if not all of the false positives should disappear. We suspect that there is a typo above and AUC should read ROC. In which case we entirely agree with the reviewer on both points and on this basis we have removed the sentence.

**4.2 ALDI calibration: Gorkha case study**

**405 – 410 This may be because longer stacks are more likely to include other landscape changes after the earthquake that disrupt the signal, such as post-seismic landslides or re-vegetation of co-seismic landslides: I guess this can be verified by looking at the series of year 2, 3, … and repeat the test of the differences year by year.**

Differentiating co-seismic and post-seismic landslides or the more general problem of identifying landslides that occurred following an unknown trigger is an interesting but much more challenging problem than the one we address here. We found that comparing results for two sequential years was difficult because of the short pre-event window. We agree that the Landsat data in GEE is well placed to address this question but our initial tests suggest the solution is not simple. Since we are here simply seeking a suggested explanation for the improved performance of short relative to long post-event windows, we would like to leave this analysis to future work.

**415 – 445: I'm sorry, I have some (maybe naive) doubts/questions related to this sub par, the topic is a bit out of my expertise, and it was also difficult to formulate my questions.**

**First, I think it would be useful to see also beta and lambda distributions to evaluate their contribution to the ratio with alpha but also their variability.**

We do not show $\beta$ and $\lambda$ distributions or dotty plots because they are redundant in the presence of the three presented datasets: dotty plots for $\alpha$, $\alpha/\beta$ and $\alpha/\lambda$. We have plotted the distributions for $\beta$ and $\lambda$ and they behave exactly as that for $\alpha$, which is expected because it is the relationship between exponents rather than

the exponents themselves which define the relative relationship between ALADIN values for different pixels. We will explain this in detail in response to the next comment.

**In general, to be honest, I'm not sure I fully understand the meaning of the ratio, or the meaning of 'controls' or 'less weight' (see 429): according to the shape of eq 4, the exponents should actually work all together to make numerically all the terms 'right' (to match the measure of goodness of fit with the manual inventory). So in the end what I don't understand is why if optimum performance always involves alpha / lambda < 1, suggests that: 1) NDVI difference should be given less weight than the more complete t-test derived probability; and 2) the additional information on pixel variability provided in the t-test does adds considerable value to ALDI for this site, and so on. Maybe my mathematical lack but according to the adopted fitting process and the type of equation (no 'interaction betw. variables e.g. dV/V(post), no physical process), I don't see the connection.**

We can show mathematically that the ratios can be substituted into the ALADIN equation:

$$ALaDIn = \begin{cases} (-dV)^\alpha \left(1 - V_{post}\right)^\beta P_t{}^\lambda, & if\ S_{post} > T_{snow}\ |\ dV < 0 \\ 0\ , & otherwise \end{cases} \quad (4)$$

By definition we can say that:

$$\beta = \alpha\ \frac{1}{\alpha:\beta}\ and\ \lambda = \alpha\ \frac{1}{\alpha:\lambda}$$

so substituting these terms:

$$ALaDIn = \begin{cases} (-dV)^\alpha \left(1 - V_{post}\right)^{\alpha \frac{1}{\alpha:\beta}} P_t{}^{\alpha \frac{1}{\alpha:\lambda}}, & if\ S_{post} > T_{snow}\ |\ dV < 0 \\ 0\ , & otherwise \end{cases}$$

The relationship between the exponents and their ratios is simpler to interpret if we take logarithms of both sides:

$$log(ALaDIn) = \alpha\ log(-dV) + \beta\ log\left(1 - V_{post}\right) + \lambda\ log(P_t) \qquad \text{for the values}$$

$$log(ALaDIn) = \alpha\ \left(log(-dV) + \frac{1}{\alpha:\beta}\ log\left(1 - V_{post}\right) + \frac{1}{\alpha:\lambda}\ log(P_t)\right) \qquad \text{for the ratios}$$

Noting that dV, $V_{post}$ and $P_t$ are all $\leq 1$, larger values of α:β result in smaller powers for the $V_{post}$ term (for a value <1 raising to a smaller power results in a larger value) thus larger α:β reflects a dominance of $V_{post}$ over dV. The same structure means that larger values of α:λ reflect a dominance of $P_t$ over dV. We have now clarified this in the ALADIN theory section (3.1), including explicit statements about what different values of the ratio mean for the relative importance of the different predictors.

In addition, we have clarified the interpretation of the ratio parameters in the calibration section (4.2) by adding the following for α:β:

*"Noting that dV, and $V_{post}$ are bounded to be <1, and that by definition $\beta=\alpha(\alpha:\beta)^{-1}$, larger values of $\alpha:\beta$ result in smaller exponents on $V_{post}$ and larger values of the term. ALADIN is thus dominated by $V_{post}$ at higher $\alpha:\beta$ ratios, and by dV at lower ratios."*

And for α:λ:

*"As explained above, ALADIN is dominated by $P_t$ at higher $\alpha:\lambda$ ratios, and by dV at lower ratios."*

**If I understand well, the a-priori distribution is a uniform distribution for the parameters of eq 4. Apparently alpha does not change very much with the different measures of goodness of fit, which might be a symptom that more runs are needed or that another choice would be more appropriate (see also the histogram).**

This insensitivity to α is expected given the form of eqn 6: α alone does not control the pattern but only the magnitude of the ALADIN surface. The interaction between the different components depends on the ratio of the exponents so it is α/β and α/λ that controls pattern, α is included for completeness but the lack of pattern

confirms our expectations. We could have replaced α with any other one of the exponents and the behaviour would have been the same. To report other exponents (e.g. β) in addition is unnecessary because they can be recovered from α and α/β and because they would show the same lack of pattern.

**Why in col b in the first two rows there are empty spaces (I think beta distribution may help here too)?**

The empty space in the dotty plot reflects a lack of ALADIN performances with that combination of parameter value α:β and performance. The reason this happens is because, for $L_{post}$ of 1-2 years and α:β of 1-5 $TPR_{diff}$ in row 2 is always greater than 0.01. This is true no matter what values are used for all other parameters! However, $L_{post}$ of 3 years even with α:β 1-5 $TPR_{diff}$ is never better than -0.01. Put another way, for this range of α:β, performances with $L_{post}$<=2 years are always better than 0.01 and those with $L_{post}$>=3 years are always worse than 0.02 leaving a gap in the point cloud. Plots for β and λ will not help here, as they will look exactly the same as that for α and as we have just shown performance is insensitive to the absolute values of these exponents though it is highly sensitive to their ratio.

**Last: when dV rather than Pt is used: Sorry I don't understand.**

Relevant text: *"dV clearly adds more value than $P_t$ for the Gorkha case study: performance worsens by ~79% for $TPR_{diff}$ when $P_t$ rather than dV is used"*

We agree that this sentence was confusing and has been removed and replaced by: *"Best performances are found for α:λ in the range 0.01-1 for $TPR_{diff}$ and 0.1-5 for AUC, suggesting that, though both layers contribute important information, dV is a stronger predictor than $P_t$ for the Gorkha case study."*

**4.3 ALDI calibration: global comparison**

**445 – 450 sensitive: are the non-sensitive parameters useless?**

Non-sensitive parameters are not useless but they are less important. If the parameter is not sensitive across its reasonable range it can be assigned any value within that range without altering model performance. However, care is needed in identifying insensitive parameters because parameter interactions can result in equifinality (the same outcome for different parameter sets) thus both equifinality and insensitivity can be responsible for a lack of shape in the upper bound of the dotty plot. For example, had we plotted alpha, beta and lambda rather than their ratios we might have concluded that the parameters were insensitive when in fact they interact in ways that introduce equifinalilty. The value of that parameter alone is not important but in combination with other parameters its value becomes important.

**450 – 455: L(post): see my previous comments. Probably there is a hidden connection between the different lengths of the time series and the processes that can cause changes in the time series in the different areas, but these connections are here not studied. I recommend reviewing this point (everywhere in the paper). I am also persuaded that these results can be influenced by the frequency sampling, and by the local characteristics, so I'm not so sure that they can be generalised (or they need further verification).**

First, we agree that there is probably a connection between the optimum time series length and the processes that can cause changes in the time series in the different areas. We do not examine this but now recognise it explicitly in the paper. Second, we also agree that optimum time-series length will be influenced both by the frequency of image acquisition, and by the local characteristics. However, we disagree that this means that they cannot be generalised instead arguing that we demonstrate that generalisation allows good (but not optimum) performance despite these controls. We can perhaps agree that further work might seek to examine how optimum parameters are related to image and landscape characteristics and to verify the suitability of ALADIN with generalised global parameters.

The original text: *"ALADIN is sensitive to $L_{post}$ for all sites but with trends that differ between sites: for Haiti and Gorkha one year is best, two years is reasonable and three years is poor; for Kashmir and Wenchuan one year is best*

*but two also gives reasonable results; for Aisen five years is best and one year is particularly poor (Figure 4 column G). "*

 is now immediately followed by this new text:

*"These site-by-site differences suggest a connection between the optimum time series length $L_{post}$, the frequency of Landsat image acquisition during the study period, and the processes that cause NDVI change at different sites (e.g., vegetation growth rates, fire, drought or post-seismic landsliding). While this does not preclude good performance of ALADIN using a global parameter set, it does imply that performance with this global parameter set will almost always be sub-optimal relative to a locally-calibrated set. However, such local calibration requires independent landslide mapping over at least part of the study area. Further work might seek to connect optimum parameters at a site with its image and landscape characteristics enabling a refinement of the parameters without the need for additional mapping."*

**460 – 465 faster and simpler: faster because the inventory can be produced after 2 years instead of 5 (but potentially limiting the quality), correct?**

The point we were making here was that the approach is faster and simpler because it does not require multiple Google Earth Engine image stack operations to generate dV, $V_{post}$ and $P_t$ layers. However, the reviewer correctly identifies a much more important benefit that we had not considered: *"an inventory to be produced after 2 years instead of 5"*; and this is a considerable improvement! The choice to examine only a 2 year post event window does potentially limit quality. We tested this but as we say at the end of the same paragraph we *"…found that the performance improvement for the global parameter set was negligible"*. We have modified the text to clarify incorporating the reviewer's very useful observation.

New text: *"In particular, the shorter the post-event window $L_{post}$ the sooner an inventory can be compiled following an earthquake. Our site-by-site calibration suggests that it is possible to find single values for these parameters that result in good performance for all study sites (Figure 4). This is the case when the cloud threshold $T_{cloud}$ is 0.5, the pre-earthquake stack length $L_{pre}$ is 5 years, and the post-earthquake stack length $L_{post}$ is 2 years thus it is reasonable to expect that an ALADIN-derived inventory can be generated after 2 years)."*

**485 out-performs it: but the competitor is now a part of the procedure to obtain the best ALDI, without a competitor, no ALDI. Does the confrontation make still sense? It sounds to me more like ALDI is a tool to improve the quality of existing inventories, or reduce the differences between inventories.**

We believe that the comparison makes sense. It is not correct that the competitor is part of the procedure to obtain the best ALaDIn, nor that without the competitor there is no ALADIN. Even in the case of locally optimised parameters, the calibration is performed on the check rather than the competitor dataset. The hold back test examines the case where no information from the test site is used to train ALADIN.

The competitor dataset influences the TPRdiff metric by defining the FPR at which the TPR for both layers will be compared. But this is an influence of the competitor on the comparison metric rather than on ALADIN itself. We have explained why we choose this metric and estimated the sensitivity of our results to other metric choices in response to earlier comments.

**4.5 Size distributions**

**550 – 555: the similarity: after resampling. So it is no more the original product.**

We agree that once the manual inventories have been resampled (i.e. rasterised and re-clustered) they are no longer the original product. We have modified the text to: 1) highlight the results of a direct comparison between size distributions from ALADIN and manual mapping; and 2) introduce the resampled manual inventories as a diagnostic tool to explain why the ALADIN size distributions differ from those in the original manually mapped inventories.

The second paragraph in this sections began: *"The size distributions derived from ALDI and those from resampled manual mapping generally exhibit a broadly similar right tail to those of the manually mapped distributions."*

This was confusing, we have modified the sentence so that this paragraph is entirely focussed on comparison between manual mapping and ALADIN: *"The ALADIN-based distributions generally exhibit a broadly similar right tail to those of the manually-mapped distributions;"*

On reflection, the previous text was too strong in its inference that ALADIN should be compared to the resampled manual maps because these maps include a resolution effect as well as a clustering effect. We have modified this to introduce the resampled inventories as a diagnostic tool and no longer claim that the resampled inventories are the most appropriate to compare to ALADIN:

*"To illustrate the role of amalgamation and censoring we convert the manual landslide maps to binary grids at 30 m resolution, using a 'majority area' rule to identify landslide-affected pixels, and perform the same connected component clustering used for ALADIN. Resampling to 30 m should result in strong censoring and some amalgamation as explained above. Re-clustering with a connected components algorithm likely results in further amalgamation. Figure 7 shows that resampling and re-clustering manually-mapped landslides transforms their area-frequency distributions removing the rollover and resulting in distributions that are very similar to those for landslide pixels classified with ALADIN. This supports our interpretation that misfit between ALADIN and manual mapping is due to censoring and amalgamation, although we are unable to determine their relative roles. Misfits due to the resolution of the Landsat and thus the classification surface are difficult to overcome whereas improvements in clustering could be more easily implemented."*

**5.1 The problem of testing against check data of only comparable quality**

**560 – 565: Sorry, I don't understand the meaning. Furthermore, what about the other 5?**

Original text: *"In the $TPR_{diff}$ cross-comparison the ALADIN classifier out-performs 8 of 14 inventories when tested against a second inventory indicating that it is more skilful than at least one of the inventories (either the check or competitor inventory)."*

We have clarified the sentence:

*"The $TPR_{diff}$ results for the five study sites show that ALADIN out-performs manual mapping in 8 of 14 inventories in terms of its ability to identify landslide-affected areas identified in a second check inventory."*

**5.2 Performance differences in manual mapping reflect inventory errors, not solely mapping errors.**

**580 severe warping: actually warping occurs when non-linear transformations and non-representative GCPs are used to orthorectify images, maybe better to say 'distortions or deformations.**

Addressed: *"warping"* changed to *"distortions"*.

**585 1 km: ok, but there is a clear mistake somewhere, co-registration / orthorectification processes are not so bad.**

We agree that there is a mistake somewhere but we point out that this is not an isolated mistake, we have found many examples of shifts and distortions in the mapping (though the example we highlight is useful because it is so clear). The examples are not limited to a particular pair of intentories or even to a particular site but appear to be characteristic of the landslide inventories that we have compared. The distortions appear to be complex (rather than only manifesting as simple offsets) and vary over relatively small (tens of km) length scales. For some study areas (e.g. Wenchuan) we find that one manual map agrees closely with ALADIN in some

areas while the other disagrees, then in other areas the roles are reversed with the second map agreeing closely and the first distorted and/or offset.

We have found limited discussion in the literature of the accuracy with which images were georeferenced prior to landslide mapping. However, Williams et al. (2018), who do discuss this, highlight the difficulties of georeferencing oblique satellite imagery in the high relief landscapes in which large numbers of coseismic landslides typically occur.

This points to a particularly useful role that ALADIN could play in evaluating existing landslide inventories to identify areas where they may suffer distortions.

**5.3 Both agreement between manual inventories and ALDI performance differ depending on the property of interest…**

**595 – 600: a more appropriate product: sorry but I keep thinking that this is not appropriate to say because, unless a fixed set of parameters is used (still, the manual inventories are somehow part of the process) in future earthquakes, ALDI needs the manual inventory in the flow chart. So, the competitor is used to get closer to the reference.**

We believe it is appropriate to say that ALDIN is a more appropriate product for the specific task of deciding whether the majority of a 30 m pixel is affected by coseismic landsliding. In this paper we show (using the holdback test) that in 57% of cases, without any information from any of the manual mapping undertaken at the study site, ALADIN performs as well or better than the competitor inventory at reproducing the test inventory. This holdback test was designed to examine whether a parameter set trained on existing inventories could be used to develop new inventories. Thus we can agree that a set of inventories was necessary to train ALADIN but now that it is trained it can label 30 m pixels as landslide affected (or not) as skilfully as manual mapping without the need for any new mapping to be undertaken at the site.

**Furthermore, hazard mapping needs an accurate definition of the size, so I don't think ALDI can be considered as the best solution in this case.**

While we agree that landslide size should play an important role in defining hazard we do not know of any existing hazard maps that account for landslide size in their assessment. In addition we know of very few landslide hazard models (and no co-seismic models) that either make predictions about landslide size or use observed landslide size in their predictive process. Thus given the current state of the art in landslide hazard mapping and modelling we disagree that accurate definition of landslide size is necessary because the information if available could not currently be used. We now discuss this in Section 5.4:

*"Current approaches to train and test landslide prediction models (including hazard and susceptibility models) almost exclusively use pixel-based information on landslide presence or absence rather than information about the size or shape of a landslide at a particular location (see Bellugi et al. (2015) for an exception). For such applications, skilful identification of landslide-affected pixels is the sole requirement. Our results suggest that the ALADIN landslide inventory would be an appropriate product to use in these cases as it is better than at least one of the manual inventories in four of the five case studies (Table 3)."*

**General comments/curiosities:**

**Does still make sense to keep Haiti in the general model?**

We retain parameter sets from Haiti because we think it reflects conditions that could be encountered both in terms of numbers of images and topographic and land cover conditions.

**Back to some of my previous comments, it seems to me that the frequency of acquisition can be relevant.**

Please see our response to the previous comment on frequency of acquisition.

**Stripe problems: please, do see one of my previous comments (I have the feeling that I see parallel lines in many ALDI products.**

You are correct that there are stripes in the ALADIN surface introduced by the scanline correction failure of Landsat 7. We now highlight these in our description and discussion of the ALADIN surfaces (see response to earlier comment).

**6 Conclusions: to be further verified**

We agree that, as with most papers, these conclusions could be further verified in future work but also believe that our conclusions are adequately supported by the results in the paper.

---

## Author Comment (AC2)

Thank you for your thoughtful comments which have helped us to considerably improve the manuscript. We hope that the responses below, together with the modifications in the manuscript address all your concerns. We have responded to each of your comments (in bold) below.

**1.NDVI based differencing approach is not new which authors also clarified in the method section. For novelty part, author have incorporated the cloud score, NDSI and temperature into the existing method. While I agree that the incorporation of NDSI and cloud score is necessary in snow covered areas (here, Gorkha), but in other areas such as in Haiti, this make things complicated. In previous studies, it has been found that vegetation recovery in earthquake affected areas take minimum 2 (Kashmir case) to more than 10 years (Wenchuan). Thus, cloud free composites of either Landsat8 or Sentinel2 images within the first or second year of event can easily be prepared in GEE platform, and should be used in cases other than Gorkha. This essentially makes two different algorithms, but I believe things will be less complicated.**

We agree that: 1) NDVI differencing is not new; 2) there are some cases where snow is not a concern; 3) cloud free composites can easily be prepared in Google Earth Engine. However, we disagree that: a) it is less complicated to present two different algorithms; and b) our algorithm converges on a comparison of before and after cloud free composites. Finally we note that even this comparison of pre- and post-event composites would require decisions about stack lengths to generate each composite.

Therefore we argue that our findings are novel not only in presenting a new algorithm (albeit following closely from previous work (e.g. Behling et al., 2014; 2016; Marc et al., 2019; Scheip and Wegmann, 2021) in a way that is outlined in Sections 1 and 3.1); but also, in identifying 'behavioural' (i.e. well performing) parameter values for the key parameters that must be defined for this type of analysis; and in demonstrating that even an algorithm as simple as the one we present here can identify landslide affected pixels with comparable skill to manual mapping.

**2.Further, to improve the performance of NDVI based difference approach in areas such as in Haiti, I would suggest authors to take minimum NDVI approach rather than average NDVI (i.e., minimum NDVI of the pixel of interest in the last, say, 5 years preceding the event). This approach will make sure that fresh barren surface caused by landslides have lower NDVI values than pre-event, and can be easily detectable and also helps in reduce the false positives.**

This is a good suggestion. Developing a minimum NDVI based algorithm could be a fruitful avenue for future research. There is certainly a rational theoretical basis for such an algorithm and it would be interesting to compare the two approaches but doing so would involve developing a second algorithm. This would take the paper in an entirely different direction and (we feel) broaden its scope beyond that which is tractable for a single paper. Thus we do not pursue it here.

**3. Although the authors have validated their method with manually delineated landslides, readers would like to know where the new approach stands when compared with other automated approaches such as HazeMapper (Scheip and Wegmann, 2020), supervised classification or machine learning techniques. These should be incorporated in discussion section.**

We have added a new section (5.5 in the discussion) comparing ALADIN to these alternative approaches:

*"5.5 Comparison to other automated detection methods*

*Automated detection of landslides typically relies on vegetation change detection and involves either generating indices of surface disturbance from which landslides can be manually identified (e.g. Scheip and Wegmann, 2020), or performing a supervised classification (e.g. Barlow et al., 2003; Behling et al., 2014; 2016; Prakash et al., 2020).*

*A recent example of automated surface disturbance detection, HazMapper (Scheip and Wegmann, 2020), uses similar image data (Landsat) and the same platform (Google Earth Engine) as ALADIN, but for a different purpose and using different functions to combine and transform the imagery. HazMapper is designed to generate a qualitative metric for surface change rather than a landslide-specific mapping tool. As a result, the approach does not mask snow-covered areas in case these are of interest for a user's particular application. The approach is simpler than that of ALADIN in that HazMapper calculates the NDVI difference only, rather than accounting for post-event NDVI, seasonal variability and noise in the NDVI signal for each pixel. It is currently only*

*applied to Landsat 7 onwards and only for individual sensors, rather than combining images from multiple Landsat sensors. This limits the events that can be examined to those occurring after 1999. However, results from HazMapper for the same study periods examined here show a good qualitative agreement with the ALADIN results. The similarity in approach, using stacks of Landsat imagery before and after a suspected trigger event, means that the two approaches will likely have many of the same strengths (e.g., the accurate georeferencing of Landsat imagery) and limitations (e.g., the coarse resolution of Landsat imagery and long wait times required to generate the post-event stack).*

*Alternative approaches to landslide detection that involve supervised classification typically rely on machine learning (e.g. Prakash et al., 2020) or clustering methods (e.g. Barlow et al., 2003; Behling et al., 2014; 2016). These more complex approaches are compatible with the data and platforms that we use here. Although we have taken a simpler approach, the classification surfaces generated by ALADIN could be coupled with modern machine learning approaches to improve ALADIN's landslide detection skill. However, our results also highlight an important potential limitation to the use of supervised learning for landslide detection in general. Given the very severe disagreement between manually-mapped landslide inventories, any supervised learning method will have a very high risk of propagating gross errors into the classifier unless the training inventory is precisely co-located with the imagery used by the classifier. ALADIN could help improve existing supervised classification efforts by providing additional well-referenced landslide inventories, or by correcting existing ones."*

**4. Among all the inventories applied in this study, the Wang et al, 2019 (Hokkaido case) is the most recent one, and is mapped from 3 m Planet imageries. There is one more inventory available for Hokkaido case (see Dou et al. 2020) which was mapped from aerial images (less than 1 m). I would like to see this results in table 3.**

We could continue to add inventories indefinitely but chose to stop at five study sites. We feel that this is sufficient. Hokkaido complicates the analysis because it opens the possibility of a Sentinel based analysis. This would require re-calibration and a more complete introduction to the properties of the Sentinel satellite and we feel that this is out of scope for the current study. We have used datasets from USGS sciencebase throughout our quantitative analysis in order to ensure consistent and traceable analysis. We include the Hokkaido dataset as an exception for illustrative purposes but do not use it in our quantitative analysis.

**5. More explanation is needed on how the ALDI pixel are converted to landslide objects. I can see that in Kashmir, Aisen and Wenchuan cases, the large landslides identified by ALDI are more than manual methods (Fig. 7). Comment on this.**

We now explain that the continuous ALADIN index is thresholded to generate the same FPR as the comparison inventory:

*"For manually-mapped inventories this information is generally captured automatically since landslides are mapped as discrete objects rather than on a pixel-by-pixel basis. However, automated classifiers like ALADIN require additional steps to convert a continuous pixel-based classification surface to a set of landslide objects. First, we generate a binary prediction of landslide presence or absence by thresholding the ALADIN classification surface to match the manually-mapped FPR, as described above."*

We then explain that landslide pixels identified by ALADIN are converted to landslide objects based on a connected components clustering:

*"Second, we convert the binary landslide map to a set of landslide objects by identifying connected components at the 30 m resolution of the Landsat imagery (Haralick and Shapiro, 1992). This connected components clustering is one of the simplest of many possible clustering algorithms."*

We have also added reference to specific examples in Fig. 7 in the results section to describe and explain the increased frequency of large landslides in the ALADIN-based distributions:

*"However, the ALADIN-based distributions, are clearly different from those derived from manual mapping, they lack: 1) the roll-over at small areas (in all cases, Figure 7a-e); 2) the positive curvature to the right tail (particularly clear for Haiti, Figure 7d); and 3) the roll-off at very large areas (resulting in oversampling of landslides >$10^5$ $m^2$ for Wenchuan, Figure 7c)."*

and

*"These differences can be explained in terms of amalgamation and censoring. Amalgamation of multiple neighbouring landslides increases the frequency of large landslides, fattening the right tail (Marc and Hovius, 2015); and in some cases considerably increasing the size of the largest landslide (e.g. Aisen and Wenchuan, Figure 7b-c)."*

**Minor comments**

1. **Xu et al 2014 inventory is having serious problem. It would be better if authors used the Fan et al., 2018 inventory for Wenchuan.**

We have used datasets available at USGS sciencebase throughout our quantitative analysis in order to ensure consistent and traceable analysis. We are not aware of an open access version of the Fan et al., 2018 inventory. However, we note that: 1) the Xu et al. (2014) inventory was collected as part of a study published in a peer reviewed journal; 2) we have not found clear evidence in the literature that the Xu et al. (2014) inventory is more seriously problematic than other inventories for Wenchuan in particular or such that it is not reflective of co-seismic inventories in general; and 3) we find that Xu et al. (2014) is consistent with ALADIN than Li et al. (2014) in some parts of the study area while the converse is true in other parts. Thus we conclude that Xu et al. (2014) is likely to contain mapping errors and that the same is true of Li et al. (2014) but that these errors are a fair reflection of the current state of the art in co-seismic landslide mapping.

**2. Figure quality should be improved, A grey background or another color would hel to distinguish the ALDI 0 from positive ALDI values.**

Thank you, this is useful. We have now sought to add a light green background to distinguish ALADIN 0 from other values. We chose to avoid grey because we already use grey to indicate areas that were masked during manual mapping. We have also sought to improve the quality of the maps by: 1) removing the grey transparencies in the full study area maps to make the ALADIN pattern easier to see at that scale; and 2) simplifying the mask layer in Figure 2 so that all masks are shown in grey to enable introduction of an additional colour to represent ALADIN=0.

**3. Delete the Xu et al inventory from Fig 8. The offset in Xu et al inventory make the visualization difficult.**

In Fig 8 the Xu et al. (2014) inventory is actually more consistent with ALADIN (and thus we argue more likely to be correct) than Li et al. (2014). Therefore it is difficult to argue for a removal of the Xu et al data without independent evidence of issues with the dataset. We have argued above that both Xu et al. (2014) and Li et al. (2014) inventories should be retained in our analysis and if that is the case then we feel that both should be displayed in Fig 8. Indeed, the striking offset between the two manually mapped inventories is one of the major points that we hope the reader will take from this figure.

**4.Title can be a more relaxed one than current one. Automated mapping still have problem of delineating source and accumulation zones. Further, in automated method, separating two landslide in adjacent slopes (special case – Hokkaido) is difficult.**

This is a point of agreement between both reviewers and while we suspect that this may reflect the dominance of the manual mapping paradigm in this field we have sought to relax the title, removing the claim that automated detection out-performs manual mapping, to reflect their concerns.

*"Automated determination of landslide locations after large trigger events: advantages and disadvantages compared to manual mapping"*

We have also added to the discussion section to expand on the limitations of ALADIN relative to manual mapping (Section 5.3) and to clarify the domains in which one out performs the other (Section 5.4).

**5. Shorten the paper a little for better reading**

We have shortened our description of the methods which was identified by R1 as a section with scope for shortening. We have also thoroughly edited the paper for clarity and readability. However, unfortunately we suspect the paper has expanded very slightly overall as a result of revisions to address comments from both reviewers. We are happy to consider further reductions in length, especially if the reviewer can provide more specific guidance.